# Pay for Hints, Not Answers: LLM Shepherding for Cost-Efficient Inference

## Abstract

Large Language Models (LLMs) deliver state-of-the-art performance on complex reasoning tasks, but their inference costs limit deployment at scale. Small Language Models (SLMs) offer dramatic cost savings yet lag substantially in accuracy. Existing approaches—routing and cascading—treat the LLM as an all-or-nothing resource: either the query bypasses the LLM entirely, or the LLM generates a complete response at full cost. We introduce LLM Shepherding, a framework that requests only a short prefix (a hint) from the LLM and provides it to SLM. This simple mechanism is surprisingly effective for math and coding tasks: even hints comprising 10–30% of the full LLM response improve SLM accuracy significantly. Shepherding generalizes both routing and cascading, and it achieves lower cost under oracle decision-making. We develop a two-stage predictor that jointly determines whether a hint is needed and how many tokens to request. On the widely-used mathematical reasoning (GSM8K, CNK12) and code generation (HumanEval, MBPP) benchmarks, Shepherding reduces costs by 42–94% relative to LLM-only inference. Compared to state-of-the-art routing and cascading baselines, shepherding delivers up to $2.8\times$ cost reduction while matching accuracy. To our knowledge, this is the first work to exploit token-level budget control for SLM-LLM collaboration. Code is available at https://anonymous.4open.science/r/LLM_Sheperding-77C3/.

## 1. Introduction

Large Language Models (LLMs) deliver state-of-the-art performance on complex reasoning tasks, but their inference costs limit deployment at scale. Meanwhile, the emergence of highly capable open-source Small Language Models (SLMs) has created new opportunities for cost-efficient inference—whether on edge devices, private servers, or shared datacenters. SLMs offer clear advantages: lower latency, improved privacy, and significantly smaller monetary and energy costs. However, for tasks involving logic and mathematics, the quality gap between SLMs and frontier LLMs (e.g., GPT, Gemini, Claude) remains substantial (Subramanian et al., 2025). Users and organizations thus face a difficult trade-off: accept inferior response quality to reduce costs, or pay a premium for LLM access to maintain accuracy.

This cost-quality trade-off motivates a central question: *How can we substantially improve SLM output quality while minimizing reliance on expensive LLMs?* The question is especially pressing in resource-constrained settings—edge devices with limited connectivity, latency-sensitive applications, and cost-conscious deployments—where every LLM call carries significant overhead. The goal is not merely faster inference, but fundamentally less LLM computation for a given quality target.

Two complementary paradigms have emerged to address this challenge: routing and cascading (Behera et al., 2025). *Routing* (Lu et al., 2024; Ding et al., 2024; Ong et al., 2025) uses a learned classifier to direct each query to exactly one model—either the SLM handles the query entirely, or it is forwarded to the LLM based on query complexity. *Cascading* (Chen et al., 2024; Aggarwal et al., 2024; Gupta et al., 2024) takes a sequential approach: the SLM attempts a response first, and the LLM is invoked only if the SLM response fails a confidence or correctness check.

We observe that both routing and cascading share a fundamental limitation: they treat the LLM as an all-or-nothing resource. Either the query bypasses the LLM entirely, or the LLM generates a complete response, incurring the full LLM inference cost. This binary view overlooks a key opportunity: *most LLM service providers allow users to specify* `max_new_tokens`, *a user-defined limit for the number of output tokens*. This capability can be strategically leveraged to reduce LLM costs while simultaneously improving SLM answer quality.

We propose **LLM Shepherding**, a novel framework that complements routing and cascading by enabling *partial*

[1]Anonymous Institution, Anonymous City, Anonymous Region, Anonymous Country. Correspondence to: Anonymous Author <anon.email@domain.com>.

Preliminary work. Under review by the International Conference on Machine Learning (ICML). Do not distribute.

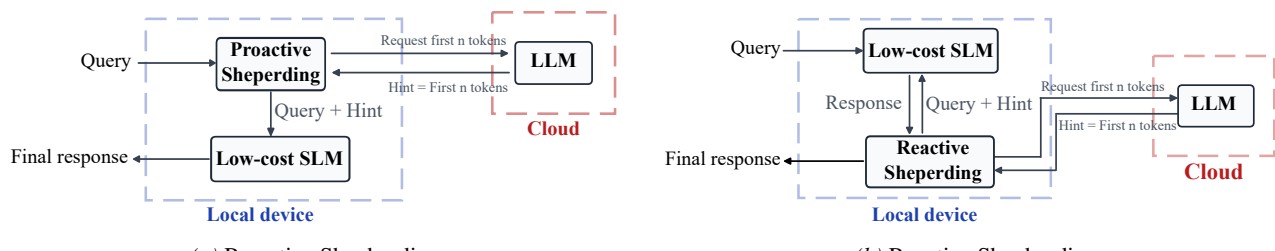

*(a)* Proactive Shepherding        *(b)* Reactive Shepherding

*Figure 1.* Two modes of LLM Shepherding. Both modes request only the first $n$ tokens from the LLM as a hint, which the SLM uses to output the response. (a) Proactive: hint decision made upfront. (b) Reactive: hint requested only after SLM failure.

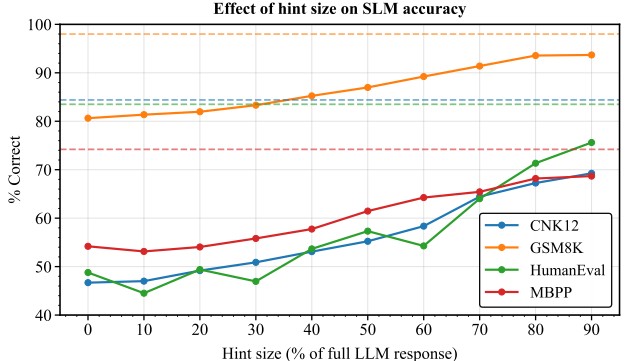

*Figure 2.* SLM (Llama-3.2-3B-Instruct) accuracy as a function of LLM (Llama-3.3-70B-Versatile) hint size. Even small hints (10–30% of the full LLM response) yield substantial accuracy gains, with diminishing returns beyond 60%. This motivates shepherding: requesting hints rather than full LLM responses. Dashed lines indicate the LLM accuracy for each dataset.

**LLM assistance.** Rather than having the LLM produce complete responses, shepherding requests the LLM to output only a short prefix – first $n$ tokens of the LLM response – called a *hint*. The hint is then concatenated with the query and is given to SLM for the final response. To validate whether LLM hints can benefit SLM inference, we evaluate on four widely-used benchmarks spanning mathematical reasoning—GSM8K (Cobbe et al., 2021) and CNK12 (OpenR1, 2025)—and code generation—HumanEval (Chen, 2021) and MBPP (Austin et al., 2021). We use Llama-3.2-3B-Instruct as the SLM and Llama-3.3-70B-Versatile as the LLM (Dubey et al., 2024). As shown in Figure 2, even small hints substantially improve SLM accuracy across all datasets.

As illustrated in Figure 1, we instantiate two modes of shepherding: 1) **Proactive Shepherding** (Figure 1a) is a *routing-based* shepherding, where a learned router directly predicts whether a hint is needed and how many tokens to request, and 2) **Reactive Shepherding** (Figure 1b) is a *cascading-based* shepherding, where the features of the SLM response(s) are used for cascading decision and hint prediction. Observe that proactive shepherding reduces to routing at the extremes: hint size zero (SLM only) or maxi-

mal (full LLM response). Similarly, reactive shepherding reduces to cascading when a complete LLM output is requested upon SLM failure. Shepherding thus generalizes both paradigms. By operating in the intermediate regime – requesting hints rather than full responses – shepherding theoretically achieves lower costs than either approach.

Realizing this vision in practice presents substantial challenges. First, the 'minimum' hint size required by different queries has a heavy-tailed distribution: on GSM8K, 80% of queries need no hint, while the rest span 10–90% of the full LLM response. Second, the benefit of additional tokens is non-smooth—critical information may appear at specific token boundaries, causing abrupt quality transitions. Third, the relationship between hint length and SLM accuracy is non-monotonic; excessive hints can over-constrain SLM generation, degrading performance even when shorter hints succeed (cf. HumanEval in Figure 2). These challenges motivate our two-stage prediction framework that jointly determines whether a hint is needed and how many tokens to request.

**Contributions:**

- We formalize LLM Shepherding and prove it achieves lower cost than routing and cascading under oracle decision-making (Section 2).

- We develop a two-stage prediction model that jointly predicts whether a query requires a hint and how many tokens to request, addressing key challenges including non-monotonic quality response and heavy-tailed hint distributions (Section 3).

- We empirically evaluate shepherding against state-of-the-art routing strategies (RouteLLM (Ong et al., 2025), GraphRouter (Feng et al., 2025)) and cascading strategies (FrugalGPT (Chen et al., 2024), ABC (Kolawole et al., 2025)) on mathematical reasoning (GSM8K, CNK12) and code generation (HumanEval, MBPP) benchmarks. Reactive Shepherding achieves the highest Accuracy-per-Cost Efficiency (ACE) on all datasets, delivering $2\times$ greater cost reduction than routing baselines on challenging multilingual queries (CNK12) and

$2.8\times$ greater cost reduction than cascading baselines on cross-domain transfer (HumanEval)—the latter without code-specific training (Section 4).

## 2. Problem Formulation

### 2.1. System and Notation

Let $\mathcal{Q}$ denote the set of all possible queries and let $q \in \mathcal{Q}$ denote an individual query. Let $\mathcal{X}$ denote a finite vocabulary of tokens and $\mathcal{X}^*$ denote the set of all finite token sequences over $\mathcal{X}$. We consider two language models: an open-source SLM $h_s : \mathcal{Q} \to \mathcal{X}^*$ deployed on a local device, such as a smartphone, laptop, or local server, and an LLM $h_l : \mathcal{Q} \to \mathcal{X}^*$ accessible through an LLM API.

For any token sequence $x \in \mathcal{X}^*$, we use $|x|$ to denote its size (number of tokens). For two sequences $x, y \in \mathcal{X}^*$, we denote their concatenation as $x \oplus y$. Given a sequence $x$ and an integer $n \leq |x|$, we write $x_{[1:n]}$ to denote the prefix of $x$ consisting of its first $n$ tokens. We use $\mathbb{1}[\cdot]$ for the standard indicator function.

We define a quality function $\phi : \mathcal{Q} \times \mathcal{X}^* \to [0,1]$ that measures the quality of a response $r \in \mathcal{X}^*$ to a query $q \in \mathcal{Q}$. Higher values indicate better response quality. In practice, $\phi$ may be instantiated as an exact-match accuracy metric (for tasks with ground-truth answers), a semantic similarity score, or a learned reward model. We define a quality threshold $\tau \in (0,1]$ and $I_s(q) = \mathbb{1}_{[\phi(q,h_s(q)) \geq \tau]}$ to indicate whether the SLM alone produces a satisfactory response.

### 2.2. Hints and Shepherding Policy

The central concept in LLM shepherding is the *hint*, defined as a prefix of the LLM's response provided to the SLM to guide its generation. Formally, for a query $q$ and hint size $n \geq 0$, the hint is given by:

$$\text{hint}(q,n) = h_l(q)_{[1:n]}$$

The hint-augmented SLM response is then:

$$h_s^{(n)}(q) = h_s\big(q \oplus \text{hint}(q,n)\big)$$

where the SLM generates a response conditioned on the original query concatenated with the hint prefix.

**Shepherding Policy.** A *shepherding policy* $\pi : \mathcal{Q} \to \{0, 1, \ldots, N_{\max}\}$ maps each query $q$ to a hint size $n = \pi(q)$, where $N_{\max}$ is the maximum output token limit of the LLM. When $n = 0$, the SLM responds without assistance; when $n > 0$, the system requests an $n$-token hint before SLM completion.

For each query $q$, we define the *minimum hint size* as the smallest number of LLM tokens required for the SLM to produce a satisfactory response:

$$n^*(q) = \min\left\{ n \in \mathbb{Z}_{\geq 0} : \phi\big(q, h_s^{(n)}(q)\big) \geq \tau \right\} \quad (1)$$

The *Oracle Shepherding Policy* $\pi^*$ maps each query $q$ to $n^*(q)$.

### 2.3. Cost Model and Evaluation Metrics

We adopt a token-based cost model consistent with commercial LLM API pricing. Let $c_l^{\text{in}}$ and $c_l^{\text{out}}$ denote the cost per input token and output token for the LLM, respectively. Similarly, let $c_s^{\text{in}}$ and $c_s^{\text{out}}$ denote the corresponding costs for the SLM. For open-source SLMs deployed on a local device, we have $c_s^{\text{in}} = c_s^{\text{out}} = 0$.

The cost of shepherding a query $q$ with hint size $n > 0$ is:

$$c_{\text{shep}}(q,n) = |q|c_l^{\text{in}} + nc_l^{\text{out}} + (|q| + n)c_s^{\text{in}} + |h_s^{(n)}(q)|c_s^{\text{out}}.$$

To compare the cost-performance trade-offs across different strategies/policies, we introduce a normalized efficiency metric called *Accuracy-per-Cost Efficiency* (ACE). Let $A_l$ denote the accuracy of the remote large model with total cost $C_l = \sum_{q \in \mathcal{Q}} c_l(q)$, and $A_s$ denote the accuracy of the on-device small model with cost zero (assuming $c_s^{\text{in}} = c_s^{\text{out}} = 0$). For a policy $\pi$ with accuracy $A_\pi$ and total cost $C_\pi$, we define *Accuracy-Cost Efficiency*:

$$\text{ACE}(\pi) = \frac{(A_\pi - A_s)/(A_l - A_s)}{C_\pi/C_l},$$

which captures accuracy gain per unit cost relative to the SLM and LLM baselines. Note that we exclude the SLM-only policy for ACE calculation because a scenario in which the SLM possesses sufficient standalone capability would render the shepherding or any SLM-LLM collaboration framework redundant.

For a fair comparison between the strategies, we also present the minimum cost achieved by the strategies while guaranteeing a target quality level. This is formulated in the following problem.

**Problem.** Given a query distribution $\mathcal{D}$ and a target expected quality $\tau$, find a policy $\pi^*$ that minimizes expected cost while achieving the quality target:

$$\begin{aligned} \min_{\pi} \quad & \mathbb{E}\left[c_{\text{shep}}(q, \pi(q))\right] \\ \text{s.t.} \quad & \mathbb{E}\left[\phi\big(q, h_s^{(\pi(q))}(q)\big)\right] \geq \tau. \end{aligned} \quad (2)$$

### 2.4. Cost Analysis (Oracle)

We analyze the monetary cost of routing, cascading, and shepherding under oracle (optimal) decision-making. Note that routing incurs either the full SLM cost (when the SLM

suffices) or the full LLM cost (when escalation is needed), but never both. Cascading always incurs the SLM cost first; if the SLM fails, it additionally incurs the full LLM cost, potentially paying for both models. Shepherding incurs only the SLM cost when no hint is needed ($n^*(q) = 0$); otherwise, it incurs (i) the LLM input cost for processing the query, (ii) the LLM output cost for generating $n^*(q)$ hint tokens, and (iii) the SLM cost for processing the augmented input. Thus, we have the following proposition.

**Proposition 2.1** (Oracle Costs). *Under oracle decision-making, the monetary cost per query $q \in \mathcal{Q}$ for routing, cascading, and shepherding is given by:*

$$c^*_{route}(q) = (|q|c^{in}_s + |h_s(q)|c^{out}_s)I_s(q)$$
$$+ (1 - I_s(q))(|q|c^{in}_l + |h_l(q)|c^{out}_l)$$
$$c^*_{casc}(q) = |q|c^{in}_s + |h_s(q)|c^{out}_s + (1 - I_s(q))(|q|c^{in}_l + |h_l(q)|c^{out}_l)$$
$$c^*_{shep}(q) = |q|\mathbb{1}_{[n^*(q)>0]}c^{in}_l + n^*(q)c^{out}_l$$
$$+ (|q| + n^*(q))c^{in}_s + |h_s^{(n^*(q))}(q)|c^{out}_s$$

*where $h_s^{(n)}(q) = h_s(q \oplus hint(q,n))$ denotes the SLM's response when augmented with an $n$-token hint.*

For open-source SLMs deployed on a local device or on private infrastructure, we have $c^{in}_s = c^{out}_s = 0$. Under this assumption, we have the following corollary (proof in Appendix A).

**Corollary 2.2.** *If $c^{in}_s = c^{out}_s = 0$, then for all queries $q \in \mathcal{Q}$:*

$$c^*_{shep}(q) \leq c^*_{route}(q) = c^*_{casc}(q).$$

The cost advantage of shepherding arises from replacing the full LLM output cost $|h_l(q)|c^{out}_l$ with the partial hint cost $n^*(q)c^{out}_l$. Since LLM output tokens are typically the dominant cost factor (with $c^{out}_l > c^{in}_l$ for most commercial APIs), even modest reductions in output token count yield significant savings.

*Remark* 2.3. The cost expressions in Proposition 2.1 and Corollary 2.2 apply directly to proactive shepherding. Reactive shepherding incurs an additional SLM inference cost; however, when $c^{in}_s = c^{out}_s = 0$, the oracle costs are identical, and Corollary 2.2 holds for both variants.

## 3. Training the Shepherding System

Our goal is to learn a policy $\pi_\theta : \mathcal{Q} \to \mathbb{Z}_{\geq 0}$, that predicts the minimum hint size $n^*(q)$. However, predicting $n^*(q)$ directly is challenging for several reasons: 1) **Heavy-tailed hint sizes:** $n^*(q)$ varies significantly across queries – on GSM8K, 80.6% of queries require no hint at all, while the remaining 19.4% span a wide range from 10% to 90% of the full LLM response (see Appendix D for a detailed discussion), 2) **Non-smooth benefit:** A small change in $n$ can

abruptly determine whether the SLM crosses the quality threshold $\tau$, as critical information (e.g., a key constraint or intermediate step) may appear at specific token boundaries, 3) **Non-monotonic quality:** The quality function $\phi(q, h_s^{(n)}(q))$ may not increase monotonically with $n$ (cf. Fig. 2). Excessive hint tokens can over-constrain the SLM, degrading performance even when shorter hints succeed, and 4) **Tokenization sensitivity:** The same semantic content yields different token counts depending on the model and query phrasing, making direct regression to absolute token counts brittle.

We address the above challenges by adopting a two-step policy design: we first predict whether any hint is needed (i.e., whether $n^*(q) > 0$), and only then predict the required hint size conditioned on needing a hint. Concretely, we train a *binary classifier* $y(q) = \mathbb{1}_{[n^*(q)>0]}$ and a *regressor* for $n^*(q)$ on the subset of queries with $n^*(q) > 0$.

### 3.1. Training Data and Supervision Signals

To train the Shepherding model, we construct supervision labels $n^*(q)$ by evaluating candidate hints at discretized hint sizes. For each query $q$, we generate hints at percentage sizes $p \in \{0, 10, 20, \ldots, 90\}$ of the full LLM response length $|h_l(q)|$, computing $n_p = \lfloor \frac{p}{100}|h_l(q)| \rfloor$ and querying the LLM with max_new_tokens $= n_p$ to obtain $hint(q, n_p) = h_l^{(n_p)}(q)$. We then evaluate each hint by running the SLM on the augmented prompt and record the minimal sufficient size:

$$n^*(q) = \min \left\{ n_p : \phi\big(q, h_s(q \oplus hint(q, n_p))\big) \geq \tau \right\}.$$

If no partial hint meets the threshold, we set $n^*(q) = |h_l(q)|$ and use the full LLM response. We provide a detailed discussion of the labeling procedure, discretization granularity, and design rationale in Appendix B.1. We filter samples where no hint size improves SLM accuracy, and the LLM response is also incorrect; see Appendix C for details.

Recall that $y(q) = \mathbb{1}_{[n^*(q)>0]}$. The log-transformed hint size is given by $r(q) = \log(1 + n^*(q))$ for queries requiring hints. The model produces logit $\widehat{y}(q)$ for hint/no-hint classification and prediction $\widehat{r}(q)$ for the hint size.

### 3.2. Model Architecture

We use DeBERTa-v3-large (He et al., 2023) as the text encoder. Let $\mathbf{h}_{CLS}(q)$ denote the [CLS] embedding of query $q$. Let $\mathbf{f}(q)$ denote the input feature vector. We pass it through a Multi-layer Perceptron (MLP) and obtain the transformed numeric features $\mathbf{g}(q) = \text{MLP}(\mathbf{f}(q))$. We construct the fused representation $\mathbf{u}(q) = [\mathbf{h}_{CLS}(q); \mathbf{g}(q)]$. Using $\mathbf{u}(q)$, the Shepherding system produces:

$$\widehat{y}(q) = \text{Head}_{hint}(\mathbf{u}(q)) \in \mathbb{R},$$

$$\widehat{r}(q) = \text{Head}_{\text{size}}(\mathbf{u}(q)) \in \mathbb{R},$$

where $\widehat{y}(q)$ is the logit for the hint indicator $y(q)$ and $\widehat{r}(q)$ predicts $r(q)$. Here, $\text{Head}_{\text{hint}}$ and $\text{Head}_{\text{size}}$ are lightweight prediction heads (small feed-forward networks) attached to the shared fused representation $\mathbf{u}(q)$: $\text{Head}_{\text{hint}}$ maps $\mathbf{u}(q)$ to a single scalar logit for binary classification (hint-needed vs. no-hint), while $\text{Head}_{\text{size}}$ maps $\mathbf{u}(q)$ to a single scalar for the regression target $r(q)$. To reduce variance in the hint-indicator logits, we apply multi-sample dropout and average logits across multiple passes (Inoue, 2019).

### 3.3. Training Objective

We optimize a joint objective over the training set. For the hint indicator, we use binary cross-entropy on the logit $\widehat{y}(q)$:

$$\mathcal{L}_{\text{hint}} = -\Big[ y(q) \log \sigma\big(\widehat{y}(q)\big) + \big(1 - y(q)\big) \log\Big(1 - \sigma\big(\widehat{y}(q)\big)\Big)\Big].$$

We address class imbalance via minibatch balancing.

For the hint size, we train only on positive examples $\mathcal{Q}^+ = \{q \in \mathcal{Q} \mid y(q) = 1\}$ using Smooth L1 (Huber) loss (Huber, 1964):

$$\mathcal{L}_{\text{size}} = \frac{1}{|\mathcal{Q}^+|} \sum_{q \in \mathcal{Q}^+} \text{SmoothL1}\big(\widehat{r}(q) - r(q)\big),$$

which penalizes the regression residual on the log-transformed target and is more robust to outliers than $\ell_2$ loss. Intuitively, $\mathcal{L}_{\text{size}}$ is computed only when a hint is needed ($y(q) = 1$), since the size is undefined for no-hint queries. The total loss is $\mathcal{L} = \lambda \mathcal{L}_{\text{hint}} + (1 - \lambda)\mathcal{L}_{\text{size}}$, where $\lambda \in [0, 1]$ is the relative weight parameter. Additional optimization details, including the use of AdamW, EMA, class balancing via weighted sampling, and policy calibration procedures, are provided in Appendix B.2.

### 3.4. Inference: Mapping Predictions to Decisions

At test time, we first compute the hint probability from the binary classification head: $\mathbb{P}(q) = \sigma(\widehat{y}(q))$, where $\sigma(\cdot)$ is the sigmoid function. We then use a threshold $\alpha \in [0, 1]$ to obtain the binary hint decision $\mathbb{1}_{[\mathbb{P}(q) \geq \alpha]}$. The predicted hint size is then defined as:

$$\pi_\theta(q) = \begin{cases} 0, & \text{if } \mathbb{P}(q) < \alpha, \\ \lfloor \text{clip}(\exp(\widehat{r}(q)) - 1, 0, N_{\max}) \rceil, & \text{if } \mathbb{P}(q) \geq \alpha, \end{cases}$$

where $\lfloor x \rceil$ denotes rounding $x$ to the nearest integer, $\text{clip}(x, a, b) = \max(a, \min(x, b))$ clamps values to the interval $[a, b]$, and $N_{\max}$ is the maximum LLM output token limit. The parameter $\theta$ encompasses all learned model components (backbone encoder, MLP, and prediction heads).

**Hint Threshold.** For queries not requiring hints ($\pi_\theta(q) = 0$), we return the direct SLM output $h_s(q)$. For queries

requiring hints ($\pi_\theta(q) > 0$), we introduce a hint size threshold $\eta_{\text{hint}}$ to further distinguish between low-complexity and high-complexity queries:

- If $0 < \pi_\theta(q) \leq \eta_{\text{hint}}$, we query the LLM for a hint by setting max_new_tokens $= \pi_\theta(q)$ and obtain $\text{hint}(q, \pi_\theta(q)) = h_l^{(\pi_\theta(q))}(q)$, and return the shepherded response $h_s(q \oplus \text{hint}(q, \pi_\theta(q)))$;

- If $\pi_\theta(q) > \eta_{\text{hint}}$, the query is deemed high-complexity and routed directly to LLM, returning $h_l(q)$ without invoking the SLM.

This threshold-based routing captures the intuition that queries requiring extensive hints may be better served by the LLM's full reasoning capacity, avoiding the overhead of generating a lengthy hint only to invoke the SLM. The threshold $\eta_{\text{hint}}$ is a hyperparameter tuned on the validation set to balance cost and accuracy.

### 3.5. Reactive Shepherding: Cascading Decision

So far, we have described the shepherding model that is used in proactive shepherding. For reactive shepherding, we add a cascading decision module before the shepherding model. Further, we also incorporate features from SLM's responses to train the shepherding model.

We make the cascading decision based on the agreement between SLM's responses – a simple, training-free technique demonstrated to be effective in (Kolawole et al., 2025). Specifically, for each query we run the SLM $K$ times independently with temperature sampling ($T = 0.3$) to generate candidate answers $\{A_1, \ldots, A_K\}$ and associated predictive entropies $\{e_1, \ldots, e_K\}$. We use stochastic sampling rather than deterministic decoding to obtain diversity in the responses, which enables us to measure the SLM's epistemic uncertainty through response variation and predictive entropy. We set $K = 3$ in our experiments as a practical balance between signal quality and computational cost. Since we work with math and coding datasets, the agreement between the responses is based on exact matches in final answers. If at least $k$ out of $K$ answers agree, we output the agreed SLM answer directly; otherwise, we invoke the Shepherding model. The choice of $k$ reflects a precision-cost trade-off: $k = 3$ (unanimous agreement) is more conservative, reducing false negatives at the expense of higher routing rates and LLM costs, while $k = 2$ (majority voting) is more aggressive, allowing more queries to bypass the LLM but risking more errors in direct SLM outputs. We explore both settings in our experiments (Section 4).

Given multiple responses from the SLM, we compute the following features and incorporate them in training our shepherding model.

- **Average entropy:** $\bar{e}(q) = \frac{1}{K} \sum_{i=1}^{K} e_i$, capturing the model's overall uncertainty;

- **Average output length:** $\bar{L}_{\text{out}}(q) = \frac{1}{K} \sum_{i=1}^{K} |A_i|$, reflecting the typical response complexity;

- **Query length:** $|q|$, measured in tokens via the backbone model's tokenizer.

We use token-level length features rather than character-level measurements because tokenization aligns with the models' internal representations and provides a more semantically meaningful proxy for reasoning complexity, longer outputs typically indicate multi-step solutions requiring more intermediate reasoning. Token-based features also enable efficient batched processing without additional tokenization overhead at inference time.

These features are standardized and fused with the text representation through a lightweight MLP. Intuitively, higher entropy indicates greater uncertainty in the SLM's responses, suggesting that a hint is necessary. The output length statistics provide additional context regarding the verbosity required for the query.

## 4. Performance Comparison

In this section, we compare the cost and accuracy of Proactive and Reactive Shepherding strategies with state-of-the-art LLM routing and cascading strategies. The routing baselines include RouteLLM (Ong et al., 2025) and GraphRouter (Feng et al., 2025), while the cascading baselines include FrugalGPT (Chen et al., 2024) and ABC (Kolawole et al., 2025). We use the vanilla Llama-3.2-3B-Instruct (without finetuning) as the SLM, deployed on a local server with an NVIDIA RTX 5090 GPU. For the LLM, we query Llama-3.3-70B-Versatile via the Groq API at $0.59 per 1M input tokens and $0.79 per 1M output tokens.

### 4.1. Datasets and Experimental Setup

**Datasets.** We evaluate all methods on four benchmarks spanning mathematical reasoning and code generation:

**(a) GSM8K** (Cobbe et al., 2021) is a grade-school mathematics dataset containing 7,473 training problems and 1,319 test problems. We use 776 questions for testing and the remaining 543 for validation. Each question requires multi-step arithmetic reasoning and includes a natural language solution and a numerical answer.

**(b) CNK12** is a Chinese K-12 mathematics benchmark derived from the OpenR1-Math-220k dataset (OpenR1, 2025), covering diverse topics including algebra, geometry, and calculus across elementary through high school levels. We randomly sample 21,471 problems from the full dataset,

partitioned into 18,251 for training, 1,073 for validation, and 2,147 for testing.

**(c) HumanEval** (Chen, 2021) is a code generation benchmark with 164 programming problems. Each problem includes a function signature, docstring, body, and multiple unit tests for verification. We evaluate this benchmark using the model trained on GSM8K without any code-specific training to demonstrate the generalization capability of our approach.

**(d) MBPP** (Austin et al., 2021) is the Mostly Basic Python Problems benchmark containing 974 crowd-sourced entry-level programming tasks, each with three automated test cases for solution verification. We evaluate MBPP in a zero-shot manner without any dataset-specific modifications.

**Inference and Evaluation.** Test sets contain no pre-computed hints. At inference time, our policy predicts a hint size $\pi_\theta(q)$; when a hint is requested, we query the LLM with max_new_tokens set to $\pi_\theta(q)$, obtaining a truncated prefix that is appended to the original prompt for SLM completion. All generations use temperature $0.3$ and top-$p$ $0.95$, detailed prompt templates are provided in Appendix E.

For mathematical benchmarks (GSM8K, CNK12), we extract the final numerical answer and compare against ground truth. For coding benchmarks (HumanEval, MBPP), we execute generated solutions and verify that all unit tests pass. To obtain robust accuracy estimates under sampling variability, we perform 7 independent trials per query and apply majority voting.

### 4.2. Performance Analysis

Tables 1–4 summarize results across four benchmarks. For the baseline routing and cascading strategies, we use default parameter settings released by the authors in their respective GitHub repositories. For Proactive and Reactive Shepherding, we present the values for $\eta_{\text{hint}}$, $\alpha$, and $K$ in Appendix F. Observe that Reactive Shepherding achieves the highest ACE on all datasets, ranging from 1.25 (CNK12) to 2.78 (MBPP), while reducing costs by 42–94% relative to LLM-only inference.

*Comparison with routing.* Routing baselines (RouteLLM, GraphRouter) struggle when query difficulty varies widely. On CNK12, they achieve only 5–8% cost reduction compared to LLM, whereas Reactive Shepherding delivers 42%—a 2$\times$ improvement at comparable accuracy. The gap is smaller on GSM8K, where most queries are SLM-solvable, but shepherding still matches Oracle performance ($0.034) while routing methods incur 2–3$\times$ higher cost.

*Comparison with cascading.* Cascading baselines (Frugal-GPT, ABC) achieve higher accuracy than routing but at a greater cost. On GSM8K, ABC reaches 94.9% accu-

*Table 1.* Cost–accuracy analysis on the GSM8K test set (776 questions). LLM ($0.104, 98.0%) | SLM ($0, 73.0%).

| Strategy | Cost ($) | Acc. (%) | Cost red. (%) | ACE |
|---|---|---|---|---|
| **Routing** | | | | |
| RouteLLM | 0.065 | 88.1 | 37.5 | 0.98 |
| GraphRouter | 0.041 | 82.2 | 60.6 | 0.93 |
| Proactive Shep. | 0.036 | 80.9 | 65.9 | 0.93 |
| **Cascading** | | | | |
| FrugalGPT | 0.047 | 86.7 | 54.8 | 1.21 |
| ABC | 0.048 | **94.9** | 53.7 | 1.89 |
| Reactive Shep. | **0.034** | 89.1 | **67.4** | **1.97** |
| Oracle Shep. | 0.034 | 98.0 | 67.4 | 3.10 |

*Table 2.* Cost–accuracy analysis on the CNK12 test set (2,147 questions). LLM ($0.559, 84.4%) | SLM ($0, 53.8%).

| Strategy | Cost ($) | Acc. (%) | Cost red. (%) | ACE |
|---|---|---|---|---|
| **Routing** | | | | |
| RouteLLM | 0.533 | **82.6** | 4.65 | 0.99 |
| GraphRouter | 0.513 | 81.7 | 8.2 | 0.99 |
| Proactive Shep. | 0.455 | 77.9 | 18.5 | 0.97 |
| **Cascading** | | | | |
| FrugalGPT | 0.391 | 76.1 | 30.1 | 1.04 |
| ABC | 0.470 | 81.3 | 15.8 | 1.07 |
| Reactive Shep. | **0.324** | 76.0 | **42.1** | **1.25** |
| Oracle Shep. | 0.306 | 84.4 | 45.2 | 1.83 |

*Table 3.* Cost–accuracy analysis on the HumanEval test set (164 problems). LLM ($0.023, 83.5%) | SLM ($0, 48.8%). All methods use models trained on GSM8K without code-specific training.

| Strategy | Cost ($) | Acc. (%) | Cost red. (%) | ACE |
|---|---|---|---|---|
| **Routing** | | | | |
| RouteLLM | 0.017 | 76.2 | 26.3 | 1.07 |
| GraphRouter | 0.020 | 78.0 | 14.0 | 0.98 |
| Proactive Shep. | **0.009** | 62.8 | **62.7** | 1.03 |
| **Cascading** | | | | |
| FrugalGPT | 0.021 | **82.9** | 7.0 | 1.08 |
| ABC | 0.019 | 76.2 | 15.8 | 0.94 |
| Reactive Shep. | 0.013 | 76.2 | 44.3 | **1.42** |
| Oracle Shep. | 0.014 | 83.5 | 38.6 | 1.63 |

*Table 4.* Cost–accuracy analysis on the MBPP test set (500 problems). LLM ($0.025, 74.2%) | SLM ($0, 65.2%).

| Strategy | Cost ($) | Acc. (%) | Cost red. (%) | ACE |
|---|---|---|---|---|
| **Routing** | | | | |
| RouteLLM | 0.006 | 67.0 | 75.2 | 0.81 |
| GraphRouter | 0.005 | 66.6 | 79.6 | 0.76 |
| Proactive Shep. | 0.006 | 67.0 | 76.8 | 0.86 |
| **Cascading** | | | | |
| FrugalGPT | 0.011 | **69.6** | 55.6 | 1.10 |
| ABC | 0.004 | 67.8 | 83.6 | 1.76 |
| Reactive Shep. | **0.002** | 67.2 | **93.6** | **2.78** |
| Oracle Shep. | 0.015 | 74.2 | 40.0 | 1.67 |

racy—5.8 points above Reactive Shepherding—but costs 41% more, yielding lower ACE (1.89 vs. 1.97). This illustrates a key trade-off: cascading pays for accuracy that may exceed application requirements, while shepherding optimizes cost-efficiency by requesting only partial LLM assistance.

*When the SLM is already capable.* On MBPP, the SLM baseline reaches 65.2% accuracy—only 9 points below the LLM. In this regime, Reactive Shepherding excels: it achieves 93.6% cost reduction with an ACE of 2.78, outperforming ABC by $1.6\times$. Notably, Reactive Shepherding even surpasses Oracle Shepherding in ACE (2.78 vs. 1.67). This occurs because Oracle optimizes for matching LLM accuracy (74.2%), which requires hints on many queries and incurs $7.5\times$ higher cost ($0.015 vs. $0.002). When the accuracy gap is narrow, accepting slightly lower accuracy yields a substantially better ACE trade-off that Reactive Shepherding captures.

*Cross-domain generalization.* HumanEval tests whether shepherding transfers to unseen domains: all methods use GSM8K-trained models without code-specific fine-tuning. Reactive Shepherding matches ABC's accuracy (76.2%) while achieving $2.8\times$ better cost reduction (44% vs. 16%), yielding the highest ACE (1.42). This zero-shot transfer suggests that hint allocation learns generalizable patterns of query difficulty rather than task-specific heuristics.

Shepherding's advantage stems from its ability to request partial hints (typically 10–30% of the LLM response) rather than full outputs. This granular control enables cost-accuracy trade-offs unavailable to binary routing and cascading methods.

### 4.3. Minimum Accuracy Requirement

To evaluate cost efficiency under constrained accuracy targets, we establish a threshold at 90% of the LLM's baseline accuracy for each dataset. Figure 3 compares the minimum cost required by each method to meet or exceed this threshold. Since routing and cascading baselines cannot fine-tune their cost-accuracy trade-offs as granularly as shepherding's continuous hint allocation, we set each baseline to its optimal operating point closest to meeting the accuracy threshold (see Appendix F for details).

Across all four benchmarks, Reactive Shepherding consistently achieves the lowest cost while satisfying the accuracy constraint, demonstrating that partial hints provide a more efficient operating point than binary escalation strategies.

On the mathematical reasoning tasks, the cost advantage is substantial. For GSM8K (accuracy threshold: 88.2%), Reactive Shepherding achieves 89.0% accuracy at $0.034—30% cheaper than ABC ($0.048), which overshoots the target

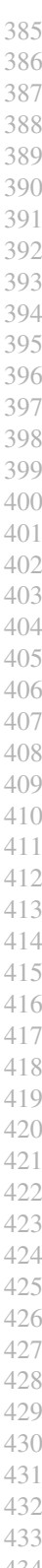

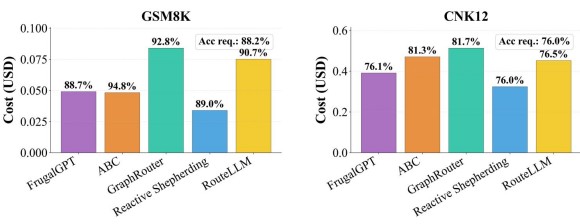

*(a)* Math datasets

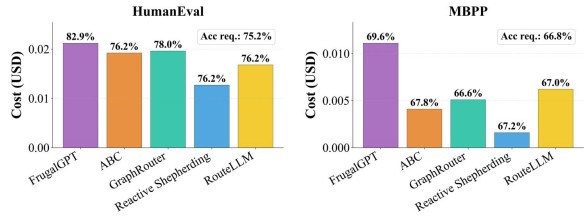

*(b)* Coding datasets

*Figure 3.* Minimum cost to achieve 90% of LLM accuracy on each dataset. Each method is tuned to its optimal operating point, which meets or exceeds the accuracy requirement (shown above bars).

at 94.8% and pays for accuracy it did not need. Frugal-GPT ($0.049, 88.7%) meets the threshold but costs 45% more, suggesting its confidence-based escalation triggers full LLM calls too aggressively. On CNK12 (threshold: 76.0%), the pattern holds: Reactive Shepherding meets the target at $0.324, while ABC and FrugalGPT incur 45% and 21% higher costs, respectively.

The advantage becomes more pronounced for code generation datasets. On HumanEval (threshold: 75.2%), Reactive Shepherding costs $0.013, 34% below ABC and 24% below RouteLLM. MBPP shows the starkest contrast: at $0.002, Reactive Shepherding is 61% cheaper than the next best method (ABC, $0.004), suggesting that code completion tasks particularly benefit from the scaffolding that partial hints provide.

We conclude that adaptive hint allocation provides superior cost efficiency when accuracy requirements are fixed, a common scenario in production deployments with service-level agreements.

### 4.4. Latency Overhead: Proactive vs Reactive

We measure inference latency on CNK12 using Llama-3.2-3B-Instruct on the NVIDIA RTX 5090 GPU. The shepherding policy adds only 7.32 ms per query—negligible compared to SLM generation (384.62 ms per run).

Proactive Shepherding makes hint decisions before SLM inference, adding minimal latency to the pipeline. Reactive Shepherding first generates three SLM responses for consensus and feature vector generation (1,163.48 ms), then invokes the decision module (7.32 ms)—a 0.6% overhead. Thus, though Reactive Shepherding achieves superior cost-accuracy trade-offs, Proactive Shepherding decision making on the local server is $159\times$ faster.

## 5. Scope and Future Directions

This work focuses on verifiable reasoning tasks, including mathematical problem-solving and code generation, where correctness can be automatically assessed. Extending shepherding to open-ended tasks like summarization or creative writing would require replacing the binary correctness function with learned reward models or human evaluation, a complementary direction we leave for future work.

Our experiments use SLM-LLM pairs from the same model family (Llama 3.2/3.3), which share a common tokenizer. An important open question is how shepherding performs when the SLM and LLM use different tokenizers or architectures (e.g., Mixtral, Qwen), where hint prefixes may not align semantically across model boundaries.

## 6. Conclusion

We have challenged the prevailing binary paradigm of routing and cascading, which treats Large Language Models (LLMs) as all-or-nothing resources. By introducing LLM Shepherding, we demonstrate that the most efficient path toward high-quality inference is not to choose between models, but to enable targeted collaboration. Our framework leverages token-level budget control, requesting only a short hint (prefix) from a frontier LLM to improve the accuracy of the Small Language Model (SLM).

Our findings establish LLM Shepherding as a new approach for cost-efficient deployment. Reactive Shepherding achieves the highest Accuracy-per-Cost Efficiency (ACE) across all benchmarks, reducing costs by 42–94% relative to LLM-only inference while delivering up to $2\times$ better cost reduction than routing baselines and $2.8\times$ better than cascading baselines at comparable accuracy. The adaptive hint allocation mechanism also demonstrates strong generalization: when trained solely on math datasets, it transfers zero-shot to code generation, matching cascading accuracy on HumanEval at a fraction of the cost. On MBPP, Reactive Shepherding even surpasses Oracle Shepherding in ACE, showing that when the SLM is already capable, partial hints outperform full LLM assistance.

## Impact Statement

This paper presents work whose goal is to advance the field of machine learning. There are many potential societal consequences of our work, none of which we feel must be specifically highlighted here.

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

## A. Proof of Corollary 2.2

Using $c_s^{\text{in}} = c_s^{\text{out}} = 0$ in Proposition 2.1, we obtain

$$c_{\text{route}}^*(q) = \left(1 - I_s(q)\right)\left(|q|c_l^{\text{in}} + |h_l(q)|c_l^{\text{out}}\right)$$

$$c_{\text{casc}}^*(q) = \left(1 - I_s(q)\right)\left(|q|c_l^{\text{in}} + |h_l(q)|c_l^{\text{out}}\right)$$

$$c_{\text{shep}}^*(q) = |q|c_l^{\text{in}} + n^*(q)c_l^{\text{out}}$$

Routing and cascading have identical Oracle costs when the SLM is free. Comparing shepherding to routing/cascading. If $I_s(q) = 1$, then $n^*(q) = 0$. Thus, all three strategies have zero cost. If $I_s(q) = 0$: Routing and cascading incur $|q|c_l^{\text{in}} + |h_l(q)|c_l^{\text{out}}$, while shepherding incurs $|q|c_l^{\text{in}} + n^*(q)c_l^{\text{out}}$. Since $n^*(q) \leq |h_l(q)|$ by definition, we have $c_{\text{shep}}^*(q) \leq c_{\text{route}}^*(q)$.

## B. Training Shepherding System: Additional Details

### B.1. Training Dataset Construction Details

This appendix provides the complete methodology for constructing the training dataset used to supervise the Shepherding model, including the discretized token budget approach and labeling procedure.

SUPERVISION VIA DISCRETIZED TOKEN BUDGETS

We obtain supervision for the hint sizes by evaluating candidate hints at multiple granularities via token-budgeted LLM calls. For each training query $q$, we first query the LLM once to obtain a reference completion $h_l(q)$ and use its LLM-token length $|h_l(q)|$ to define percentage sizes $p \in \{0, 10, 20, \ldots, 90\}$. We choose 10% increments as they provide sufficient granularity to capture the non-smooth benefit curve of hint tokens while maintaining computational tractability—finer granularities (e.g., 5% steps) would double the labeling cost with diminishing returns in identifying the minimal sufficient size, whereas coarser granularities (e.g., 25% steps) risk missing critical transition points where hints become effective.

For each $p$, we compute the target token size:

$$n_p = \left\lfloor \tfrac{p}{100}|h_l(q)| \right\rfloor$$

and then re-query the LLM by setting the `max_new_tokens` parameter to $n_p$. This parameter acts as a hard constraint on the generation length, ensuring the model outputs a completion $h_l^{(n_p)}(q)$ that conforms to the pre-defined size $n_p$.

We treat this size-constrained completion as the candidate hint $\text{hint}(q, n_p) = h_l^{(n_p)}(q)$, and evaluate it by running the SLM on $q \oplus \text{hint}(q, n_p)$ and computing $\phi\big(q, h_s(q \oplus \text{hint}(q, n_p))\big)$.

We record the discretized minimal sufficient size

$$n^*(q) = \min\left\{n_p : \phi\big(q, h_s(q \oplus \text{hint}(q, n_p))\big) \geq \tau\right\},$$

which is a grid-based approximation to the continuous optimum. The case $p = 0$ corresponds to $\text{hint}(q, 0) = \emptyset$, i.e., the router chooses not to query the LLM for any hint and directly returns the SLM output $h_s(q)$. If none of the partial sizes up to $p = 90$ meets the threshold, we set $n^*(q) = |h_l(q)|$ and use the full LLM response as the output.

RATIONALE FOR DESIGN CHOICES

**10% Granularity.** The choice of 10% increments balances labeling efficiency with precision. Each training query requires $|\{0, 10, 20, \ldots, 90\}| = 10$ LLM calls to construct the supervision signal. Finer granularities would provide marginally better approximations to $n^*(q)$ but at high computational cost—5% steps would require 20 calls per query, doubling the labeling budget. Conversely, coarser granularities (e.g., 25% steps) risk missing critical transition points where the hint becomes sufficient, particularly for queries with sharp quality transitions.

**Deterministic LLM Decoding.** For all hint-labeling LLM calls, we use deterministic decoding (`temperature= 0`, `top_p= 1`) to ensure reproducible supervision signals and eliminate variance from sampling across different training runs. This design choice prioritizes consistency in the training labels over diversity in hint content.

**Grid-based Approximation.** The discretized target $n^*(q)$ serves as a practical surrogate for a continuous optimum. While this introduces quantization error, the 10% grid is sufficient to capture meaningful differences in hint requirements across queries, and the log-transformation of the regression target (Section 3) further smooths over small discretization artifacts.

### B.2. Optimization and Implementation Details

We train with AdamW (Loshchilov & Hutter, 2019) and maintain an exponential moving average (EMA) of parameters for evaluation and checkpoint selection (Morales-Brotons et al., 2024).

**Class Balancing.** Given the potential class imbalance (i.e., when the proportion of queries requiring hints deviates from 50%), we employ `WeightedRandomSampler` during training to balance the minibatch distribution. This ensures that both positive and negative examples are adequately represented in each gradient update, yielding robust decision boundaries.

**Policy Calibration.** Post-training, we perform a grid search on the validation set to calibrate the policy for a target cost size. Specifically, given a fixed cost constraint $C_{\text{target}}$, we search over the probability threshold $\alpha$ for the binary classification decision to identify the configuration that maximizes validation accuracy while satisfying the cost constraint. This allows us to adapt the trained model to different cost-accuracy operating points without retraining.

## C. Outlier Analysis

### C.1. Outlier Generation Strategy

To identify and filter outliers in our training datasets, we employ a consensus-based approach that leverages the agreement patterns across different hint sizes. For each sample, we generate SLM responses across 11 different hint-size configurations and compare them with the ground truth. At each hint size level, we classify the outcome as an outlier if the minority class (correct or incorrect) occurs, and as normal otherwise. We then aggregate these binary outlier flags to compute $k$, the total number of outliers per sample across all hint size settings.

Samples with high outlier counts (e.g., $k \geq$ threshold) exhibit inconsistent behavior across varying hint sizes and may introduce noise during training. We therefore filter out such samples to refine the dataset quality. Figures 4 and 5 illustrate the distribution of outlier counts for the CNK12 and GSM8K datasets, respectively.

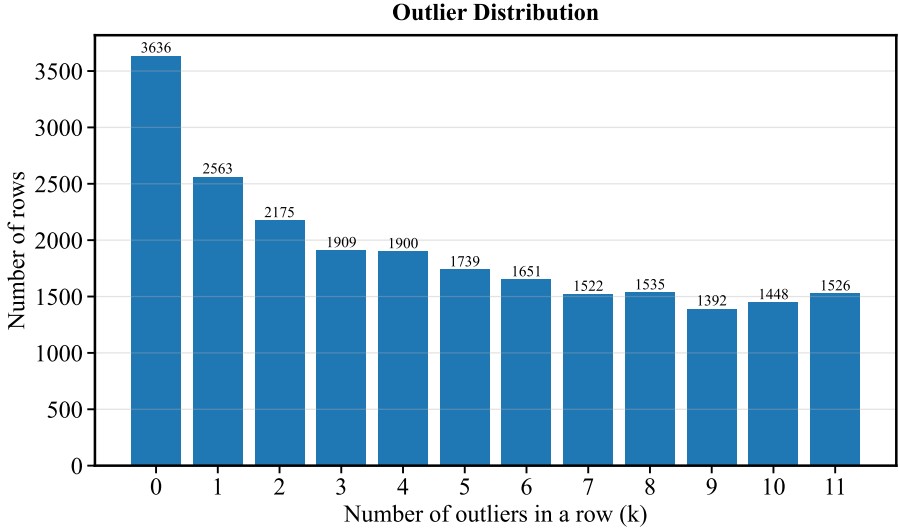

*Figure 4.* Outlier distribution for CNK12 dataset.

Note that we apply this outlier filtering strategy only to CNK12 and GSM8K, as these datasets contain sufficient training samples to justify statistical filtering. For HumanEval, which consists of only 165 problems designed specifically for evaluation purposes, we do not perform outlier analysis. Similarly, the MBPP dataset contains only 374 samples in its training set, making it impractical to remove additional samples without significantly reducing the training data size. Therefore, no outlier filtering is applied to HumanEval or MBPP.

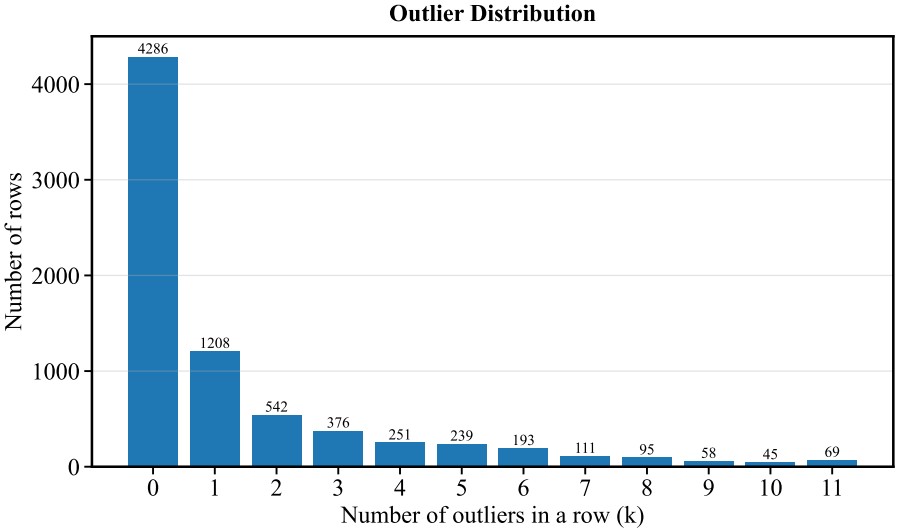

*Figure 5.* Outlier distribution for GSM8K dataset.

## D. Heavy-Tailed Distribution of Hint Sizes

As mentioned in Section 3, the distribution of required hint sizes exhibits a pronounced heavy-tailed characteristic. Figure 6 and Figure 7 illustrate this distribution across our training datasets CNK12 and GSM8K, respectively.

The distributions across both datasets demonstrate several key properties that motivate our two-stage prediction approach:

- **Dataset-dependent class imbalance:** The proportion of queries requiring hints varies significantly between datasets. GSM8K shows extreme imbalance with 80.6% (6,026 out of 7,473) requiring no hint, while CNK12 is more balanced with 46.7% (10,736 out of 22,996) requiring no hint. This dataset-dependent behavior necessitates a flexible approach that can handle varying levels of class imbalance.

- **Consistent heavy-tail structure:** Despite different absolute proportions, both datasets exhibit similar exponential decay patterns in their hint size distributions. In CNK12, the distribution decays from 13.6% (10% hints) to 1.4% (90% hints), while in GSM8K it decays from 6.6% (10% hints) to 0.4% (90% hints). The consistent decay ratio across hint sizes (approximately 0.6-0.8×) confirms the heavy-tailed nature.

- **High variance:** The standard deviation of hint sizes (computed only for queries requiring hints, i.e., $> 0\%$) is substantial in both datasets. Directly regressing on raw token counts would be highly sensitive to these large values, motivating our log-transformation strategy: $r(q) = \log(1 + \tilde{n}^*(q))$ to compress the target range.

- **Variable unsolvability rates:** The proportion of queries unsolvable even with 90% hints varies substantially: 8.7% (2,008) in CNK12 versus only 1.3% (97) in GSM8K. This indicates CNK12 contains more inherently challenging queries that exceed the SLM's capabilities even with substantial guidance. These queries are excluded from hint-size prediction because they require full LLM deployment.

- **Exponential decay characteristic:** Both datasets show clear exponential rather than linear or uniform decay. For example, in CNK12: 13.6% → 8.1% → 5.8% → 4.5%, and in GSM8K: 6.6% → 3.0% → 2.3% → 1.5%. This power-law-like behavior distinguishes the distribution from normal or uniform patterns and validates the need for specialized handling.

This heavy-tailed distribution poses significant challenges for direct regression:

1. A single regression model would struggle to accurately predict both the numerous no-hint cases (which constitute 47-81% depending on the dataset) and the rare large-hint cases (which are critical for maintaining quality on difficult queries).

*Figure 6.* Distribution of minimum required hint sizes on CNK12 dataset (22,996 queries).

2. Standard $\ell_2$ loss would be dominated by reconstruction error on the tail, potentially sacrificing accuracy on the more frequent short-hint cases.

3. The varying class balance between datasets (47-53 split in CNK12 vs 81-19 split in GSM8K) requires flexible handling via weighted sampling to ensure robust decision boundaries across different data distributions.

4. The presence of unsolvable queries (1.3-8.7% depending on dataset) necessitates explicit handling to avoid the model attempting to predict hints for inherently infeasible cases.

Our two-stage approach—first classifying whether any hint is needed, then predicting the size only for positive cases—naturally decomposes this challenging distribution into two more manageable sub-problems: a balanced binary classification (via weighted sampling that adapts to dataset-specific imbalance) and a regression on the conditional distribution $p(n^*|n^* > 0)$, which, while still heavy-tailed, excludes the dominant zero mass and focuses modeling capacity on the continuous range of hint sizes. This decomposition proves robust across datasets with varying difficulty profiles, as evidenced by the consistent performance on both the balanced CNK12 and the imbalanced GSM8K datasets.

## E. Prompting Details

This section provides detailed prompting procedures used for large-model hint generation, small-model inference, and shepherding supervision.

### E.1. Prompt Template

Across all datasets, the small or large model receives the query with the same instructions and, when available, a generated large-model hint as additional contextual input. The general prompt structure is:

> **Instruction:** Produce a final answer following the task-specific constraints.
> **Question:** {query}
> **Hint:** {generated large-model output}

When no hint is provided, the prompt is identical except that the `Hint:` line is omitted.

**GSM8K:** We use a fixed few-shot prompt consisting of two example question–answer pairs, followed by the target query. When available, a generated large-model hint is appended as additional context.

*Figure 7.* Distribution of minimum required hint sizes on GSM8K dataset (7,473 queries).

**MBPP:** The small and large models receive the function specifications, function signature, and provided test cases, along with the generated large-model hint if available. The model is instructed to generate only executable Python code, without additional explanation. The prompt structure otherwise follows the general template above.

**HumanEval:** Follows the same prompting procedure as MBPP; however, the test cases and function signature are already included in the dataset's original prompt.

**CNK12:** Follows the same prompt template as GSM8K but uses single-shot prompting without example question–answer pairs.

## F. Method Configurations for Minimum Accuracy Experiments

Table 5 details the hyperparameter settings used for each method across all datasets in the minimum accuracy requirement experiments (Section 4.3). Each baseline is tuned to achieve accuracy closest to the 90% LLM accuracy threshold on each dataset. For RouteLLM, the values shown in the table are for the routing threshold. For FrugalGPT, the values represent the confidence threshold below which queries are escalated to the LLM. GraphRouter uses either its "Performance First" (P.F.) mode, which prioritizes accuracy, or an additional bias term added to increase cost orientation. ABC uses its default configuration with $N = 2$, checking agreement between two SLM responses before escalating to the LLM. For Reactive Shepherding, $\eta_{\text{hint}}$ denotes the hint token budget; $K$ denotes the number of SLM responses used in consensus evaluation, and $\alpha$ denotes the probability threshold for the binary hint classification decision.

*Table 5.* Method configurations for minimum accuracy experiments. P.F. = Performance First mode; Bias = additive term to increase cost orientation.

| Method | GSM8K | CNK12 | HumanEval | MBPP |
|---|---|---|---|---|
| RouteLLM | 0.48 | 0.57 | 0.70 | 0.58 |
| GraphRouter | P.F. | P.F. | P.F. + $4 \times 10^{-7}$ Bias | P.F. |
| FrugalGPT | 0.0006 | 0.001 | 0.001 | 0.00016 |
| ABC | $N = 2$ | $N = 2$ | $N = 2$ | $N = 2$ |
| Reactive Shepherding | $\eta_{\text{hint}} = 58$ $K = 3, \alpha = 0.228$ | $\eta_{\text{hint}} = 60$ $K = 2, \alpha = 0.349$ | $\eta_{\text{hint}} = 110$ $K = 2, \alpha = 0.228$ | $\eta_{\text{hint}} = 130$ $K = 2, \alpha = 0.372$ |

# G. More Related Works

Existing techniques for efficient, accurate, or low-cost SLM-LLM collaborative inference include: 1) LLM routing, 2) LLM cascading, 3) speculative decoding, and 4) knowledge distillation.

## G.1. LLM Routing

LLM routing addresses cost-quality trade-offs by using a learned router to decide *a priori* which model should handle a given query. Early work by **Lu et al.** (Lu et al., 2024) introduced **Zooter**, a reward-guided routing method that distills rewards from training queries to learn a routing function, achieving top performance on 44% of tasks across 26 benchmark subsets while maintaining computational efficiency. Building on this foundation, **Hybrid-LLM** (Ding et al., 2024) proposed quality-aware query routing that dynamically trades quality for cost based on predicted query difficulty, achieving up to 40% fewer calls to large models with no quality degradation. **RouterBench** (Hu et al., 2024) subsequently established a standardized evaluation framework comprising over 405k inference outcomes from representative LLMs, enabling systematic comparison of routing strategies. **RouteLLM** (Ong et al., 2025) advanced the field by introducing a principled framework for learning routers from human preference data, demonstrating cost reductions of over 85% on MT-Bench and 35% on GSM8K while maintaining 95% of GPT-4's performance. Most recently, **GraphRouter** (Feng et al., 2025) proposed an inductive graph framework that models contextual interactions among tasks, queries, and LLMs as a heterogeneous graph, achieving at least 12.3% improvement over prior routers and enhanced generalization to unseen LLMs.

Despite these advances, routing exhibits a fundamental structural limitation: the decision is inherently binary (SLM or LLM), and when the LLM is selected, the system incurs the full inference cost. Routing does not facilitate partial reuse of LLM knowledge, rendering it insufficiently flexible for strict cost constraints.

## G.2. LLM Cascading

LLM cascading arranges models hierarchically, escalating queries to more capable models when weaker models fail quality criteria. **FrugalGPT** (Chen et al., 2024) pioneered this approach, demonstrating up to 98% cost reduction by learning which LLM combinations to use per query. **LLM-Blender** (Jiang et al., 2023) introduced an ensemble perspective, using pairwise ranking to select among candidate outputs and generative fusion to merge top candidates. **AutoMix** (Aggarwal et al., 2024) advanced cascading through few-shot self-verification and POMDP-based routing, reducing costs by over 50%. Recent work has explored both learned and training-free approaches: **C3PO** (Chen et al., 2025) provides label-free cascading with formal cost guarantees, while **Agreement-Based Cascading (ABC)** (Kolawole et al., 2025) proposes a simple, training-free strategy where escalation decisions are based on agreement among an ensemble of model responses. In our single-SLM setting, we adapt ABC by measuring consensus across multiple responses from the same SLM.

While cascading reduces average cost, it remains ill-suited for strict budget constraints: SLMs frequently fail on complex tasks, triggering full LLM inference. Each escalation requires a complete LLM forward pass, limiting cost savings when SLM reliability is low.

## G.3. Speculative Decoding

Speculative decoding (Leviathan et al., 2023; Chen et al., 2023) is a widely adopted technique for accelerating LLM inference without sacrificing output quality. The approach uses a smaller *draft model* to generate candidate token sequences, which are then verified in parallel by the larger *target model*. Tokens that match the target model's distribution are accepted, amortizing the cost of large-model inference across multiple tokens. **Medusa** (Cai et al., 2024) extends this idea by adding multiple decoding heads to the LLM itself, eliminating the need for a separate draft model. **Draft & Verify** (Zhang et al., 2024) achieves self-speculative decoding by selectively skipping intermediate layers during drafting, providing up to $2\times$ speedup without additional training. **Online Speculative Decoding** (Liu et al., 2024) continuously adapts the draft model during inference to improve token acceptance rates.

While speculative decoding and shepherding both involve collaboration between small and large models, their objectives differ fundamentally. Speculative decoding aims to *accelerate* LLM inference by parallelizing token verification, but the large model remains involved in checking every output token. In contrast, shepherding aims to *minimize* LLM usage by having the SLM produce the final response with only partial LLM guidance.

### G.4. Knowledge Distillation

Knowledge distillation (KD) (Hinton et al., 2015) transfers capabilities from a large teacher model to a smaller student model during training. Recent work has extended KD to LLMs: **MiniLLM** (Gu et al., 2024) proposes reverse KL divergence as the distillation objective, which better suits generative language models by preventing the student from overestimating low-probability regions. **Distilling Step-by-Step** (Hsieh et al., 2023) extracts chain-of-thought rationales from LLMs to train smaller models more data-efficiently, enabling a 770M-parameter model to outperform few-shot prompted 540B PaLM. Comprehensive surveys (Gou et al., 2021; Xu et al., 2024) provide extensive coverage of KD techniques for LLMs, including skill-specific and domain-specific distillation.

Knowledge distillation and shepherding are complementary but distinct. KD is a *training-time* technique: once distillation is complete, the student model operates independently at inference. Shepherding, in contrast, is an *inference-time* collaboration that dynamically injects LLM hints based on each query's difficulty. KD improves the SLM's baseline capability, while shepherding provides targeted assistance when that baseline proves insufficient.

### G.5. Positioning of LLM Shepherding

Our work differs fundamentally from all the above approaches. Unlike routing and cascading, which make binary decisions about *which* model generates the complete response, shepherding controls *how much* LLM output to request, enabling a continuous trade-off between cost and quality. Unlike speculative decoding, which accelerates LLM inference while keeping the LLM in the critical path for every token, shepherding removes the LLM from final response generation entirely. Unlike knowledge distillation, which operates during training, shepherding provides dynamic, query-specific guidance at inference time.

Shepherding complements existing approaches: proactive shepherding enhances routing by replacing full LLM calls with partial hints, reactive shepherding enhances cascading by substituting full escalations with targeted guidance, and shepherding can be combined with distilled SLMs for further improvements. To our knowledge, this is the first work to exploit token-level budget control for SLM-LLM collaboration.

