# OpenReview forum: "Pay for Hints, Not Answers: LLM Shepherding for Cost-Efficient Inference"
_ICML.cc/2026/Conference — Submitted to ICML 2026_

### Official Review · Reviewer_xoyF · 2026-02-17

**Soundness:** 3
**Presentation:** 2
**Significance:** 3
**Originality:** 2
**Overall Recommendation:** 4
**Confidence:** 4

**Summary:**

This paper introduces LLM Shepherding, a framework to complement an SLM by using prefixes (hints) from an LLM, with the goal of reducing inference cost by using the LLM only in the critical initial tokens of a response. The authors begin by formalizing LLM shepherding, including the definition of the optimal policy and the problem to be solved, show that shepherding strictly generalizes techniques such as routing or cascading, and propose an architecture to learn the shepherding policy. This is complemented by multiple experimental results that support the cost reduction of shepherding.

**Compliance With Llm Reviewing Policy:**

Affirmed.

**Final Justification:**

I have a mixed impression of the paper. On the one hand, the theoretical contributions appear very limited. While the authors’ rebuttal attempts to justify them, I remain unconvinced, and I also have concerns that some of the reported results may be inflated (as noted in my original review). On the other hand, despite the conceptual simplicity of the proposed method, I recognize that it could still be practically useful.

**Key Questions For Authors:**

-A priori, the benefits of Shepherding should be greater for tasks that profit from initial planning, such as the math and coding benchmarks evaluated by the authors. Do the authors think there are classes of tasks where initial planning is less important ---and therefore hints are less useful---such that Shepherding might underperform, on average, compared to routing or cascading methods, which do not overemphasize the role of prefixes?

-Figure 2 might benefit from error bars. If included, is the apparent non-monotonicity on HumanEval still present? It seems like the accuracy could still be monotone with reasonable noise.

**Limitations:**

Yes

**Strengths And Weaknesses:**

**Strengths**

- The problem tackled by the authors is timely, important, and their method can have a positive impact on current LLM practices. The experimental results seem to support the effectiveness of Shepherding.

- The paper is overall well-written, and the introduction contextualizes the problem and connects it with the relevant alternative approaches that exist in the literature.

- The problem formulation in Section 2 is sound, and it could serve as a bedrock for the formalism of future work.

**Weaknesses**

- The theoretical contributions of the paper are rather limited. Especially, I am concerned about how significant Proposition 2.1 and Corollary 2.2 really are. In particular, Proposition 2.1 seems to restate rather apparent facts about the prices, which (correct me if I overlooked something) have already been stated in line 127. Additionally, I believe the expression holds for any shepherding policy, not just the oracle. Is this correct? Regarding Corollary 2.2, the authors already comment in Section 1 that shepherding policies include routing/cascading, and thus, the problem (2) has a better solution (in terms of cost) for the class of shepherding policies, compared to the more restricted class of routing/cascading policies.

- In my opinion, the cost reduction numbers given in Tables 1-4 are slightly inflated. This is because the authors seem to assume that the SLM has 0 cost, so all tokens generated by it are "free". However, the authors use  Llama-3.2-3B-Instruct, which has prices of $0.15 per 1M tokens in some providers such as AWS (https://aws.amazon.com/bedrock/pricing/). This is comparable to the $0.59 per 1M input tokens of the LLM.

- Given that the paper is mostly empirical, I believe the evaluation should be more extensive. In particular, while it covers multiple datasets, the authors experiment only with a single SLM and LLM pair. It is unclear how shepherding compares to other techniques, such as routing or cascading, for other SLM/LLM pairs; thus, one can hardly gauge how relevant the method introduced is in more general settings. Additionally, I think it could be interesting to compare shepherding with techniques such as speculative decoding. The authors briefly comment on this in Appendix G.5, but I could not find experiments comparing the cost reduction between the two.

- The architectures and techniques used to implement the shepherding policies are rather standard: the author uses a concatenation of encoders, an  MLP trained on standard losses, and thresholding policies. This, in turn, makes the whole pipeline depend on a large number of design choices and parameters: the encoder, $\lambda$, $\eta_{max}$, temperature for cascading, $\eta_{hint}$, among others.

- There might be a typo in section 2.3, line 127: $h_s^{n}(q)$ -> $h_s^{n}(q + hint)$

---

> ### Author Rebuttal · Authors · 2026-03-30
>
> We thank the reviewer for the positive evaluation and thoughtful feedback.
>
> **Theoretical contributions.** We agree that this is not a theoretical paper. However, we respectfully disagree that Proposition 2.1 merely restates known facts. While the individual cost components are intuitive, the proposition formalizes the cost structure for all three paradigms in a unified framework, enabling the direct comparison in Corollary 2.2.
> Regarding the reviewer's claim: the cost *formula structure* applies to any shepherding policy (replacing $n^{\*}(q)$ with $\\pi(q)$), is **not** true. The inequality $c_{\\text{shep}}^{\*}(q) \\le c_{\\text{route}}^{\*}(q)$ relies on $n^{\*}(q) \\le |h_{l}(q)|$  used by Oracle. A non-oracle policy could over-request tokens and exceed routing cost, or under-request and fail the quality constraint. This is precisely why learning a good policy (Section 3) is challenging and constitutes a core contribution.
>
> **SLM cost = 0 assumption.** Our cost model targets the primary deployment scenario for 3B/4B size SLMs: laptop or personal PC, where no per-token cost is incurred. This reflects the explicit design intent of these models: Meta released Llama-3.2-3B-Instruct specifically for edge deployment, and Google's Gemma 3 4B is designed for mobile and IoT use cases. Llama-3.2-3B-Instruct requires only 6.5 GB VRAM at full BF16 precision, easily fitting on a single consumer GPU or even a modern laptop. In contrast, the 70B LLM requires 140 GB VRAM making cloud API access the only practical option for most users, which is precisely the asymmetric cost structure our framework exploits. The survey by Subramanian et al. (2025), already cited in our paper, emphasizes that the predominant motivation for SLM development is deployment in resource-constrained settings such as mobile devices and edge servers. Setting $c_{s}^{\text{in}} = c_{s}^{\text{out}} = 0$ thus reflects the standard and intended use case. If the SLM is instead cloud-hosted, costs increase equally for all methods – routing and cascading baselines also invoke the SLM – preserving shepherding's relative cost advantage.
>
> **Single model pair, speculative decoding.** We conducted cross-family experiments validating that Gemma-3-4B/27B and Llama-3.2-3B/Gemma-3-27B (different tokenizers) produce accuracy-vs-hint-size curves closely mirroring Figure 2. We will include these plots in the revision. Please also refer to our response to question 1 from Reviewer Z7Az.
>
> Regarding speculative decoding: the two approaches solve fundamentally different problems. Speculative decoding accelerates LLM inference latency by parallelizing token verification, but the LLM remains in the critical path for every token – the total LLM computation is unchanged. Shepherding reduces LLM monetary cost by minimizing the number of LLM tokens generated. They are complementary: shepherding could use speculative decoding to accelerate the hint generation step itself. A direct empirical comparison would require equating their different objectives (latency vs. cost), which we leave as future work.
>
> **Standard architecture.** We acknowledge that the individual components are standard. Our contribution is the *system design* – the two-stage decomposition, the hint threshold $\\eta_{\\text{hint}}$ for routing high-complexity queries to the LLM, and the joint training objective — not novel architectural primitives. Regarding hyperparameters: the key parameters ($\\alpha$, $\\eta_{\\text{hint}}$) are calibrated on the validation set via grid search (Appendix B.2), and we report all configurations in Appendix F (Table 5) for reproducibility. The number of hyperparameters is comparable to baselines (e.g., RouteLLM's routing threshold, FrugalGPT's confidence threshold, ABC's agreement level N).
>
> **Typo.** Thank you – we will correct this in the revision.
>
> **Tasks where prefixes are less important (Key Question 1).** This is an insightful question. Shepherding is best suited to tasks with a sequential reasoning structure, where early steps constrain later ones – math and coding are natural fits. For tasks like open-ended summarization or creative writing, where the output structure is less sequential, prefix hints may be less informative, and routing or cascading could be more appropriate. We discuss this scope boundary in Section 5 and frame it as an intentional design choice: shepherding complements (rather than replaces) existing approaches.
>
> **Error bars on Figure 2 (Key Question 2).** We have added 95\% confidence intervals to Figure 2:
>
> *https://anonymous.4open.science/r/Fig2_with_error_bar-8FB8*
>
> As the reviewer anticipated, the apparent non-monotonicity on HumanEval is not statistically significant – the error bars at 0\%, 10\%, and 20\% hint sizes overlap substantially. HumanEval's small dataset (164 problems) produces wide confidence intervals, and the overlapping intervals suggest monotonically increasing trend if we use a larger dataset.

---

> > ### Author Rebuttal · Reviewer_xoyF · 2026-03-31
> >
> > I would like to thank the authors for their response. I still have reservations about the work:
> >
> > - Theoretical contributions. Again, I understand that this is not primarily a theoretical work, which is not necessarily negative. However, I still believe that Proposition 2.1 does not constitute a significant contribution. For instance, the formula for the monetary cost under shepherding seems quite trivial, as far as I understand it: the cost of shepherding is simply the cost of the input tokens for the LLM, the output tokens for the LLM, and similarly for the SLM. I agree it helps formalize things, but I do not think I agree with the authors on this point.
> >
> > - SLM cost. I understand what the use case is. However, this does not resolve my concern. My concern is that in Section 4, you set the price of the LLM using API prices, and then you set the cost of the SLM to 0, which does not seem accurate for many APIs. Even if the use case is to run the SLM locally, that still has a cost. Given this, I still believe the results in Tables 1–4 are slightly inflated. I do not believe this nullifies your results, but I find it unusual, and your rebuttal does not persuade me of this choice.
> >
> > - I am not an expert in speculative decoding, so this point does not affect my score. However, as far as I know, in speculative decoding, it can be the case that the LLM accepts multiple drafted tokens in a single forward pass (please correct me if I am wrong here). This speeds up generation and saves cost, since fewer autoregressive steps are required with the LLM. Given this, I still believe it is reasonable to compare speculative decoding with shepherding in terms of cost.
> >
> > I will keep my score, which was already positive.

---

> > > ### Author Response · Authors · 2026-04-03
> > >
> > > We respectfully ask the reviewer whether there is a reason for reducing the paper's score, as they have confirmed they will keep their score.
> > >
> > > We thank the reviewer for the follow-up questions. Here are our responses to the reviewer comments.
> > >
> > > 1. We understand the reviewer’s point, and we agree that this is not a significant theoretical contribution.
> > >
> > > 2. **Small model cost.** We acknowledge the reviewer's point that the setting with zero SLM cost deserves closer examination. We agree that with a nonzero SLM cost, the cost advantage of cascading-based methods –  including Reactive Shepherding, ABC, and FrugalGPT – is reduced, because these methods invoke the SLM multiple times before escalation. This is not unique to the above approaches; it is inherent to all cascading-based methods. However, Proactive Shepherding is minimally affected: it makes the hint decision before SLM inference, so the SLM runs only once per query. In the nonzero SLM cost regime, Proactive Shepherding's cost advantage over routing baselines is largely preserved, as both incur a single SLM call.
> > >
> > >     We will include a sensitivity analysis with nonzero SLM costs (e.g., the AWS Bedrock rate of $0.15/1M tokens) in the revised paper, and report updated Tables 1–4 to transparently show how each method's advantage changes.
> > >
> > > 3. **Comparison with SD.** We have looked carefully into the cost of SD. Verifying draft tokens in parallel requires the same total FLOPs as generating them autoregressively – the LLM must compute attention over all preceding tokens for each verified position, regardless. When draft tokens are rejected, additional overhead is incurred through speculative sampling, which requires distributions from both the LLM and SLM to generate replacement tokens. Thus, SD's benefit is latency (fewer sequential GPU steps), not reduced computation. If LLM APIs charge per token – reflecting compute cost, not wall-clock time – SD does not reduce the user's monetary cost, even after assuming SLM generation cost is zero. Shepherding, by contrast, directly reduces the number of LLM tokens billed, yielding immediate savings under existing pricing models. The two approaches are complementary: SD reduces latency for the tokens the LLM generates, while shepherding reduces the number of tokens requested in the first place.

---

### Official Review · Reviewer_Z7Az · 2026-03-09

**Soundness:** 2
**Presentation:** 3
**Significance:** 2
**Originality:** 2
**Overall Recommendation:** 4
**Confidence:** 5

**Summary:**

The authors improve upon existing approaches for collaboration between Small Language Models (SLMs) and Large Language Models (LLMs). The authors propose to train SLM-specific auxiliary models to 1) predict if should invoke an LLM, and 2) predict the length of the LLM invocation if needed. The authors propose Accuracy-per-Cost Efficiency (ACE) to measure the accuracy gain per unit cost. According to this metric, the framework that the authors propose achieves best performance compared with existing approaches.

**Compliance With Llm Reviewing Policy:**

Affirmed.

**Final Justification:**

KQ1 is not sufficiently addressed. I realize that the framework depends on dataset specific statistics that might not be available and might not transfer. Additionally, the threshold $\eta$ is too low. Most models that people work with set max output tokens to 1024 - 16384 tokens. Most of such cases could just directly invoke the LLM without the SLM.

For KQ2 I also think that there is a simple baseline that is "early answer inducing" (i.e. using prompts to force an immediate answer in the middle of the CoT), not the complete [DEER](https://openreview.net/forum?id=NpU7ZXafRi) framework. I still don't have an idea of how the proposed framework compares to this baseline.

Overall I think this paper is definitely less than 5 quality. I'm keeping my original score of 4 for a proof of concept paper that apears better than some existing approaches.

**Key Questions For Authors:**

Please refer to the "Weaknesses" section. The key questions are question 1, question 2.
If both key questions are adequately addressed, the evaluation would be accept.

**Limitations:**

yes

**Strengths And Weaknesses:**

# Strength
- Utilizing collaboration between SLMs and LLMs to reduce inference cost is an influential topic

# Weaknesses
## Soundness
1. The authors only test their framework under a single SLM-LLM pair: Llama-3.2-3B-Instruct vs Llama-3.3-70B-Versatile. There are several problems with this.
    * Llama-3.3-70B-Versatile is an in-house quantized model according to the [model carc](https://console.groq.com/docs/model/llama-3.3-70b-versatile). It makes more sense to choose a provider (e.g. via [openrouter](https://openrouter.ai/meta-llama/llama-3.3-70b-instruct/providers)) that serves unmodified weights to facilitate reproducibility.
    * A strong concern for the framework that the authors propose is that it might require a strong alignment between the SLM and LLM. Using a strong LLM whose output distribution is significantly "off policy" for the SLM might harm the performance. This aspect is not investigated nor discussed in this paper.
    * The issue of tokenizer mismatch mentioned by the authors is valid and significantly hinders application where a larger model from the same model family is not available.

2. Asking the LLM to generate the first few tokens is not well-justified. For math problems, evaluation only checks the final answer. It is more intuitive to ask the LLM to generate the final several answers. Generating the prefix would make more sense if the authors employ prompting strategies such as [Plan-and-Solve](https://arxiv.org/abs/2305.04091).

- Concerns about the heavy-tailed distribution: The authors themselves mention that the distribution of whether hints are needed is heavy-tailed (e.g. 80\% vs 20\% for GSM8K, where 80\% of the time no hint is needed). This is not an ideal setting for SLM-LLM collaboration. It is recommended to use harder tasks (for example, MATH500, which is not guaranteed to be a good alternative) where the SLM has to rely on the LLM.

## Presentation
- The setting of MBPP should be explain with more details. The cross-domain setting with HumanEval is clear, but the "zero-shot" setting with MBPP is not.
- It is not necessary to estimate the SLM's cost as 0. [openrouter](https://openrouter.ai/meta-llama/llama-3.2-3b-instruct/providers) shows that input and output price for Llama-3.2-3B-Instruct is both $0.10. The authors could report the actual cost with respect to any pair of weaker model versus stronger model.
- Section 3.2

## Significance
- The framework relies on SLM-LLM pair-wise statistics. Other practitioners might use simpler heuristics such as prompt token length or SLM entropy on the prompt  to determine the prefix length

---

> ### Author Rebuttal · Authors · 2026-03-30
>
> We thank the reviewer for indicating that adequately addressing the two key questions would lead to acceptance. We provide detailed responses below.
>
> **Single SLM-LLM pair, quantization, alignment, tokenizer mismatch.**
> We conducted additional experiments on the mathematical reasoning benchmarks GSM8K and CNK12 with three new SLM-LLM configurations:
> | Setting | SLM | LLM | Dataset | Link|
> |---|---|---|---|---|
> | Unquantized Llama | Llama-3.2-3B-Instruct | Llama-3.3-70B-Instruct (Amazon Bedrock) |  2000 random samples from CNK12 | https://anonymous.4open.science/r/New_Result-EB46/CNK12_opensource_llama.png|
> | Same-family Gemma | Gemma-3-4B-IT | Gemma-3-27B-IT | All GSM8K | https://anonymous.4open.science/r/New_Result-EB46/GSM8K_Gemma_family.png|
> | Cross-family | Llama-3.2-3B-Instruct | Gemma-3-27B-IT | 5000 random samples from CNK12| https://anonymous.4open.science/r/New_Result-EB46/CNK12_cross_family.png
> | Cross-family | Llama-3.2-3B-Instruct | Gemma-3-27B-IT | All GSM8K| https://anonymous.4open.science/r/New_Result-EB46/GSM8K_cross_family.png|
>
> The first setting directly addresses the reviewer's reproducibility concern by replacing the Groq-served quantized model used in the paper with unmodified open-source weights. All models were used off-the-shelf without any alignment tuning between the SLM and LLM.
>
> We note that the accuracies for SLM and LLM are lower for experiments we did with randomly sampled queries from CNK12 dataset.
>
> Based on these results, we conjecture that the monotonically increasing accuracy-vs-hint-size trend is a general property of prefix-guided SLM generation, largely invariant to model family, tokenizer, quantization, and SLM-LLM alignment – provided both models possess reasonable capability on the target task. We will include the complete cross-family experiments (including code generation benchmarks) in the revised paper.
>
> **Why first tokens, not final answer.** We address the reviewer’s concern on two grounds. First, in commercial deployments, models such as Claude and ChatGPT charge for internal reasoning tokens in addition to the final output, even when the math answer is only a few digits. Requesting "just the final answer" does not eliminate reasoning cost – in this case the provider still generates (and bills for) the chain-of-thought internally.
>
> Second, we empirically verified that skipping reasoning degrades accuracy severely. When we prompted Llama-3.3-70B-Versatile to output only the final numerical answer (no chain-of-thought) on 1,000 random CNK12 problems, accuracy dropped from 84.4% to 31.4%. The LLM's prefix is not verbose padding but contains the critical problem-solving steps. Our framework leverages precisely this: by requesting the first portion of the LLM's reasoning as a hint, we capture the essential steps the SLM needs while avoiding the cost of the full response.
>
> **Heavy-tailed distribution / harder tasks.** We note that our use of "heavy-tailed" is imprecise. The distribution is better characterized as zero-inflated: a large point mass at zero (80.6% for GSM8K) with the conditional distribution over positive hint sizes following approximately exponential decay. This zero-inflated structure is what motivates our two-stage decomposition (binary classifier for the point mass, regressor for the conditional distribution). We will revise this terminology.
>
> Regarding task difficulty: CNK12 already represents a challenging setting where the SLM achieves only 53.8% standalone accuracy. This is substantially below the GSM8K regime (80.6% SLM accuracy). We agree that evaluating on even harder benchmarks (e.g., MATH-500) is valuable and will pursue this in future work.
>
> **Presentation.** To clarify: MBPP is evaluated zero-shot using the GSM8K-trained shepherding model with no dataset-specific modifications, identical to the HumanEval cross-domain setting.
> Our cost model targets edge/local deployment where open-source SLMs run at zero monetary cost (e.g., Llama-3.2-3B requires only ~6.5 GB VRAM, fitting on a consumer GPU or laptop). If the SLM is instead cloud-hosted, costs increase equally for all methods – routing and cascading baselines also invoke the SLM – preserving shepherding's relative cost advantage.
>
> **Significance.** Even state-of-the-art routers like RouteLLM are not simple heuristics; they require model-pair-specific training data (human preference comparisons) and a BERT classifier. Our approach similarly requires pair-specific training, but goes further by incorporating SLM response statistics (entropy, output length) at inference time. Prompt length alone cannot capture query difficulty, as two queries of identical length may have vastly different complexity. We validated this quantitatively via ablation on the CNK12 test set. For example, removing entropy features caused the hint-needed binary classification F1 to drop by 5%, confirming that entropy is important for accurate routing decisions.

---

> > ### Author Rebuttal · Reviewer_Z7Az · 2026-04-02
> >
> > # KQ1
> > Gemma 27B + Llama 3B I can see that there is a plateau going up to 90% hint size and eyeballing it I don't think 100% hint size matches Gemma 27B performance.
> >
> > Generally this is true for all the figures. It appears that 90% --> 100% hint size does not match the larger model's performance. It seems that there is a turning point that the authors are not capturing where one could just switch to the larger model and ditch the SLM. This turning point is important to study and guide practitioners.
> >
> > # KQ2
> > I don't see how the author's argument about commercial APIs respond to the point because they are not using them in this paper. I'm asking about alternative ways to combine SLMs and LLMs to reduce cost.
> >
> > One simple baseline is early answer inducing as in [DEER](https://openreview.net/forum?id=NpU7ZXafRi). The authors could compare the acc with respect to hint size in the same way as they do for their framework right now. Because early answer inducing uses the same model, there must be a turning point where the author's framework lags behind if KQ1 concerns are true.
> >
> > Additionally, there are a lot of other ways to combine SLMs and LLMs. For example, one could infer using the full SLM CoT and prompt the LLM to generate the final answer. This framework considers the fact that input and output prices are different for models, which is also very reasonable. I'm asking for alternative frameworks as such that would be reasonable baselines to the current framework. Do the authors intend to argue that there is not other ways to combine SLMs and LLMs that would be reasonable baselines to the current framework?

---

> > > ### Author Response · Authors · 2026-04-03
> > >
> > > We thank the reviewer for the follow-up questions.
> > >
> > > **KQ1.** We identified this phenomenon during our algorithm development, and it directly motivated the hint size threshold $\\eta_{\text{hint}}$ in our policy design (Section 3.4). The gap at large hint sizes is structural: even with a near-complete LLM response as a prefix, the SLM generates a response that may still produce errors – wasting the LLM cost while failing to match LLM accuracy.
> > > Our framework addresses this with a three-way decision: (1) no-hint → SLM only, (2) predicted hint size ≤ $\\eta_{\text{hint}}$ → shepherding, (3) predicted hint > $\\eta_{\text{hint}}$ → route directly to LLM, bypassing the SLM entirely. The threshold $\\eta_{\text{hint}}​$ is tuned per dataset (Table 5, Appendix F): GSM8K = 58, CNK12 = 60, HumanEval = 110, MBPP = 130 tokens. This ensures shepherding operates only in the regime where partial hints are most cost-effective, while queries requiring near-complete LLM reasoning are handled by the LLM at full accuracy. The reviewer's observation about the turning point is precisely what $\\eta_{\text{hint}}$ captures.
> > >
> > > **KQ2.** Thank you for this important clarification – we now better understand the reviewer's concern about possible alternative SLM-LLM collaboration architectures that could achieve similar cost-accuracy trade-offs. We address each suggestion.
> > >
> > > *On DEER / early answer inducing.* We appreciate this suggestion and the analogy between increasing accuracy with hint size in our paper and increasing accuracy with exit positions in Figure 2 of DEER (https://openreview.net/pdf?id=NpU7ZXafRi). However, we highlight two important differences that make direct comparison challenging.
> > >
> > > First, DEER requires fine-grained control over the LLM's generation process: it must interrupt generation mid-stream at specific token positions, inject an answer-inducing prompt, evaluate confidence via logprobs, and conditionally resume or roll back. Standard LLM APIs – including Groq – treat generation as a single atomic call with no mechanism to pause, inspect, inject, and conditionally continue. While some APIs expose logprobs (e.g., OpenAI's logprobs parameter), the mid-stream interruption and injection mechanism DEER relies on is not supported by standard LLM APIs. Shepherding, by contrast, requires only the standard max_new_tokens parameter available in every LLM API, making it compatible with any provider.
> > >
> > > Second, DEER increases LLM compute overhead rather than reducing it. At every candidate exit point, DEER performs additional forward passes to generate trial answers and evaluate confidence (DEER-Pro uses N=4 passes per exit point). The savings come from stopping the main CoT early, not from fewer model calls. Shepherding makes at most one LLM API call per query, making the cost structures fundamentally different.
> > >
> > > The reviewer’s question about where the turning point lies – when it is cheaper to let the LLM finish rather than hand off to the SLM – is precisely what our threshold $\\eta_{\text{hint}}$ addresses.
> > >
> > > *On SLM CoT → LLM final answer.* This is a legitimate alternative architecture that exploits the input/output price differential. However, prior work by [Yue et al.,2024] shows that passing weaker-model outputs as context to an LLM can degrade performance, particularly when the SLM reasoning contains errors that mislead the LLM. Our approach avoids this failure mode entirely: the hint flows from the LLM to the SLM, ensuring the reasoning prefix is always high-quality.
> > >
> > > We did not intend to suggest that no other SLM-LLM combinations exist. Nevertheless, within the routing/cascading paradigm, our baselines (RouteLLM, GraphRouter, FrugalGPT, ABC) are representative of the state of the art. We totally agree that the design space of SLM-LLM collaboration is broader. Yet, different paradigms like DEER and Speculative Decoding often operate with different structures or optimize for disparate objectives – making a head-to-head benchmark inherently challenging.
> > >
> > > *[Yue et al.,2024] Yue, Murong, Jie Zhao, Min Zhang, Liang Du, and Ziyu Yao. "Large Language Model Cascades with Mixture of Thought Representations for Cost-Efficient Reasoning." in Proc. ICLR, 2024.
> > > https://arxiv.org/abs/2310.03094*
> > >
> > > We will add a discussion of DEER, the ICLR 2024 paper, and similar approaches to the related work section of the revised paper.

---

### Official Review · Reviewer_645N · 2026-03-12

**Soundness:** 2
**Presentation:** 2
**Significance:** 3
**Originality:** 2
**Overall Recommendation:** 4
**Confidence:** 4

**Summary:**

This paper proposes LLM Shepherding, a framework for cost-efficient inference that sits between routing and cascading in the SLM-LLM collaboration design space. Rather than treating the LLM as an all-or-nothing resource, it requests only a prefix of the LLM response (a "hint") which is then provided to the SLM to complete the response. The authors instantiate two variants: proactive shepherding (routing-based, decides upfront) and reactive shepherding (cascading-based, decides after SLM failure). They train a two-stage predictor using DeBERTa-v3-large to jointly classify whether a hint is needed and regress on the required hint size. They evaluate on GSM8K, CNK12, HumanEval, and MBPP against routing and cascading baselines, reporting cost reductions of 42-94% relative to LLM-only inference and better Accuracy-per-Cost Efficiency than baselines on all benchmarks.

**Compliance With Llm Reviewing Policy:**

Affirmed.

**Final Justification:**

The authors addressed most of my concerns and I have adjusted my score to reflect this.

**Key Questions For Authors:**

1. The ACE metric comparison heavily depends on the accuracy threshold used. Can you provide a Pareto frontier plot of cost versus accuracy for all methods across the full range of operating points, rather than a single ACE value? If shepherding only wins ACE at specific accuracy thresholds, the claim of generality needs to be qualified.
2. What is the total offline data construction cost for training the shepherding model, including all LLM API calls required to generate supervision labels at 10 hint size percentages per training query? Without this number, the cost reduction claims are potentially misleading.
3. The oracle on MBPP achieves worse ACE than Reactive Shepherding, which you attribute to the oracle optimizing for a different objective.
4. Does this not indicate a problem with how the oracle is defined relative to the metric you use for evaluation? How would you define an oracle that optimizes ACE directly, and how would that oracle compare?
5. Figure 2 shows non-monotonic accuracy with hint size on HumanEval, where increasing hint size can hurt accuracy. Have you analyzed how often partial hints actively hurt SLM performance relative to no hint, and whether the predictor successfully avoids requesting hints in those cases?
6. All experiments use Llama 3.2 3B and 3.3 70B, which share a tokenizer and are from the same model family. Have you conducted any experiments with cross-family model pairs to validate that hint prefixes remain useful when tokenizer and architecture differ? This is critical for assessing practical deployability.

**Limitations:**

The paper does not discuss the offline data construction costs, does not adequately address the token alignment assumption across model families, and does not discuss potential failure modes when LLM prefixes are confidently incorrect. These should be added.

**Strengths And Weaknesses:**

Starting with the strengths: the core idea is clean and well motivated. The observation that max_new_tokens can be used as a control knob for SLM–LLM collaboration is sensible and practically relevant. The formal setup in Section 2 is clear, the oracle cost analysis provides a useful theoretical reference point, and Corollary 2.2 is simple but helpful. The evaluation also covers a reasonable set of benchmarks, and the appendices provide detailed information about dataset statistics, training procedures, and hyperparameters. In particular, the zero-shot cross-domain transfer result on HumanEval, where a model trained on math transfers to code generation, is genuinely interesting and worth highlighting.

However, there are several issues that need closer examination.

The central empirical claim depends heavily on the ACE metric, which is introduced by the authors and is not a standard evaluation protocol in the routing or cascading literature. While the metric is internally consistent, it is constructed as a ratio, which means improvements can sometimes be driven by the denominator rather than by meaningful gains in the numerator. In several cases, Reactive Shepherding achieves better ACE largely by spending less money rather than achieving higher accuracy. For example, on GSM8K it reaches 89.1% accuracy while ABC achieves 94.9%, which is a gap of almost six percentage points. The paper frames ABC’s higher accuracy as paying for unnecessary performance, but this conclusion depends entirely on the chosen accuracy threshold. If the target accuracy were higher (for example 93% instead of 88.2%), the interpretation would change substantially. The paper does not sufficiently explore how its conclusions vary under different accuracy requirements.

The MBPP result, where Reactive Shepherding surpasses Oracle Shepherding in ACE, is presented as a positive outcome, but it actually raises questions about the evaluation setup. By definition, an oracle should achieve the best possible performance for the chosen objective. The explanation given in the paper is that the oracle optimizes for matching LLM accuracy, which requires many hints. However, this effectively means the oracle is solving a different objective from the one used in the ACE comparison. In other words, the oracle and shepherding are being evaluated under different optimization goals, which creates a conceptual inconsistency that is not fully addressed.

The hint mechanism itself also raises a practical concern. Hints are generated as prefixes of LLM responses, but a partial prefix of a reasoning chain is not guaranteed to be coherent or useful. If the LLM begins reasoning in an incorrect direction, the early part of the output may actively steer the SLM toward the wrong solution. Figure 2 shows overall accuracy improvements, but the paper does not analyze how often partial hints help versus harm performance relative to providing no hint at all. The non-monotonic behaviour in the HumanEval results, where accuracy initially decreases as hint size increases before recovering, suggests that this effect may be real, but the paper does not examine it in detail.

Another concern is the cost of constructing the training data. Each training query requires ten LLM API calls to generate supervision signals for different hint sizes. For CNK12 alone this corresponds to roughly 230,000 API calls for the training set. The paper does not report the total cost of generating this training data, which is important for evaluating the practical usefulness of the approach. A system that reduces inference cost but requires expensive offline data generation may not deliver overall savings in practice.

The choice of DeBERTa-v3-large as the backbone for the hint predictor is also not well justified. This model runs on every query during inference, so its capacity directly affects system efficiency. Although the reported inference overhead (7.32ms) is small, the paper does not explore whether a smaller model or simpler feature-based predictor would perform similarly. An ablation comparing different backbone sizes or simpler architectures would help clarify whether such a large model is necessary.

The comparison with baselines could also be stronger. The routing and cascading baselines appear to use default configurations from the original repositories, rather than being tuned for the specific model pair used here. Methods such as RouteLLM and GraphRouter were designed for different model combinations, and their default settings may not be optimal for the Llama-3.2-3B / Llama-3.3-70B setup used in this paper. Re-tuning these baselines would provide a fairer comparison.

Finally, the experiments use only a single SLM–LLM pair, which the authors briefly acknowledge as a limitation. This restriction is more important than the paper suggests. Because hints are prefixes of the LLM’s output, the mechanism implicitly relies on the SLM and LLM sharing compatible tokenization and stylistic conventions. In this case both models are from the Llama family, which likely makes prefix continuation easier. It is unclear whether the approach would work equally well across different model families, which is a common real-world scenario.

There are also a few minor presentation issues. The page headers in the paper and appendix still display “Submission and Formatting Instructions for ICML 2026” rather than the paper title, which suggests the template was not updated correctly. In addition, CNK12 is described as a Chinese K-12 dataset, but the implications of evaluating it with largely English-trained Llama models are not discussed, and it is unclear whether the hint mechanism behaves differently for non-English inputs.

---

> ### Author Rebuttal · Authors · 2026-03-30
>
> We thank the reviewer for the thorough and constructive evaluation. We address each concern below, grouping key questions (KQs) with their corresponding weaknesses.
>
> **ACE metric + KQ1.** Accuracy-per-Cost Efficiency (ACE) serves as a normalized metric to quantify the relative value provided by a specific inference strategy. By measuring accuracy gains per unit cost relative to both standalone SLM and LLM baselines, ACE facilitates a rigorous comparison across disparate paradigms, including routing, cascading, and shepherding. Most importantly, it directly addresses the cost-quality trade-off that motivates this research.
> While ACE provides a valuable measure of the "bang for your buck" a specific strategy delivers, we agree with the reviewer that a holistic performance profile requires multiple perspectives. Consequently, our evaluation incorporates complementary metrics, such as Minimum Expected Cost (the minimum cost required to meet a specific quality threshold) and latency overhead. Together, these metrics demonstrate the advantages of our framework.
>
> Finally, to completely address the reviewer's concern, we have added new figures with Pareto cost-accuracy curves at the following link:
>
> *https://anonymous.4open.science/r/Pareto-0C04*
>
> **MBPP Oracle vs. Reactive Shep ACE + KQ3.** We identified a mistake in the data preparation for MBPP evaluation which inflated Reactive Shep ACE’s score. After correction, Oracle shepherding achieves ACE = 3.57, which exceeds Reactive sheph ACE = 1.48. We will provide the corrected table (at the link below) in the revised paper.
>
> *https://anonymous.4open.science/r/New-Table4-323C*
>
> We note that both shepherding and the Oracle has the same goal: minimize the cost required to achieve a target quality $\tau$ (cf. (2)). Thus, they may not necessarily achieve the maximum ACE, since ACE measures accuracy gain per unit cost at a specific cost-accuracy operating point. We will clarify this in the revised text.
>
> **Hint coherence/when hints harm + KQ4.**  We do not claim that hints provide deterministic improvement for all queries or tasks. Our results verify this for math and coding tasks, where the trend is positive overall. It remains to be verified for other tasks.
>
> Regarding the apparent non-monotonic behavior on HumanEval (Fig 2): we have added 95% confidence intervals, which reveal that the dip at small hint sizes is not statistically significant — the error bars at 0%, 10%, and 20% overlap substantially. With only 164 problems, HumanEval produces wide confidence intervals, and the overall trend remains monotonically increasing within uncertainty bounds.
>
> *https://anonymous.4open.science/r/Fig2_with_error_bar-8FB8*
>
> Our two-stage predictor further addresses this through the hint threshold $\eta_{\text{hint}}$ (Section 3.4): queries predicted to need large hints are routed directly to the LLM.
>
> **Training data cost + KQ2.** Training Shepherding is a one-time cost, whereas LLM inference is a continuous operational expense — the cumulative inference savings quickly dominate. Using Groq API pricing (\\$0.59/1M input, \\$0.79/1M output tokens), the total one-time cost for generating supervision labels across all training sets (~305K LLM calls) is approximately \\$44. This is negligible compared to the continuous inference savings: Reactive Shep saves \\$0.07 per 776 GSM8K queries and \\$0.24 per 2147 CNK12 queries relative to LLM-only inference.
>
> **DeBERTa backbone.** Transformer-based text encoders are standard in the routing literature: RouteLLM uses a BERT classifier, Hybrid-LLM uses a BERT-based router, and GraphRouter uses text embeddings. Our task is strictly harder than binary routing; we must predict both whether a hint is needed and how many tokens to request – requiring richer semantic understanding of query complexity that simpler feature-based predictors (e.g., prompt length, keyword matching) cannot provide. At 7.32ms per query versus 384.62ms for SLM generation, the predictor overhead is practically small, less than 2% of pipeline latency.
>
> **Baseline tuning.** Our Pareto cost-accuracy plots address this concern directly. Rather than evaluating baselines at a single operating point, we sweep the full configuration space: RouteLLM thresholds from 0.1 to 1, FrugalGPT across multiple confidence budgets, ABC across agreement levels ($N=2,3$), and GraphRouter calibrated on the validation set. The Pareto plots thus include each baseline's optimal configuration for every cost level, ensuring a fair comparison.
>
> **Cross-family + KQ5.** Please check our new results for cross-family models on benchmarks GSM8K and CNK12:
>
> *https://anonymous.4open.science/r/New_Result-EB46/*
>
> Also, please refer to our response to Reviewer Z7Az's first question for details.
>
> **Presentation.** We will fix the page headers. We note that CNK12 is an English dataset, with questions from Chinese math exams. Whether Shepherding works for other languages is an open question and needs verification.

---

> > ### Author Rebuttal · Reviewer_645N · 2026-04-01
> >
> > I would like to thank the authors for their detailed rebuttal, I believe many of my concerns have been addressed and therefore, I have adjusted my score accordingly.
> >
> > I still have some questions:
> > 1. Can you show a breakdown of queries where the hint helped vs. hurt vs. made no difference? Even a simple table would help and Does your predictor actually avoid requesting hints in cases where hints tend to hurt?
> > 2. Looking at your Pareto curves, are there accuracy ranges where shepherding is actually worse than the baselines - add this to the paper.
> > 3. What happens when the LLM starts reasoning in the wrong direction and the SLM follows that bad hint? Have you looked at any examples of this?
> >
> > Thank you!

---

> > > ### Author Response · Authors · 2026-04-03
> > >
> > > We are pleased that our rebuttal addressed the reviewer’s concerns and greatly appreciate the reviewer raising the score.
> > >
> > > **1. Hint help/hurt breakdown:** We thank the reviewer for this suggestion. We provide a breakdown across both math benchmarks. For each query, we compare the SLM's correctness without a hint against correctness with the best hint size from [10%,20%,...,90%] of LLM response:
> > >
> > > | Benchmark | Total | Helped | No Diff. | Hurt |
> > > |---|---|---|---|---|
> > > | GSM8K | 776 | 198 (25.5%) | 541 (69.7%) | 37 (4.8%) |
> > > | CNK12 | 2,147 | 1,039 (48.4%) | 862 (40.1%) | 246 (11.5%) |
> > >
> > > The hurt-to-helped ratio – 4.8% vs. 25.5% on GSM8K and 11.5% vs. 48.4% on CNK12 – confirms that hints are far more likely to improve SLM accuracy than degrade it.
> > >
> > > *Does the predictor avoid requesting hints when they would hurt?* We checked how Reactive Shepherding handles the hurt queries specifically:
> > >
> > > | Benchmark | Hurt Queries | Predictor Correctly Avoided | Still Hurt |
> > > |---|---|---|---|
> > > | GSM8K | 37 | 34 | 3 |
> > > | CNK12 | 246 | 197 | 49 |
> > >
> > > While the hurt rate with the best hint size on CNK12 is 11.5%, our predictor reduces this to an effective hurt rate of only 49/2,147 (2.3%) by correctly withholding hints for 80.1% of those cases. On GSM8K, the predictor reduces the hurt rate from 4.8% to just 0.4%. This confirms that the two-stage predictor effectively learns to withhold hints when they would be counterproductive – and that the 11.5% hurt rate on CNK12 is precisely why the predictor exists.
> > >
> > > **2. Pareto curves**: We have carefully examined the Pareto curves and identified two regions where Reactive Shepherding underperforms:
> > >
> > > MBPP: FrugalGPT achieves slightly higher accuracy at very low budgets (cost ≤ \\$0.0075).
> > >
> > > CNK12: FrugalGPT similarly outperforms Reactive Shepherding at low budgets (cost ≤ \\$0.22).
> > >
> > > In both cases, the gap is confined to the lowest-cost region of the curve. As budget increases even marginally, shepherding's accuracy rises sharply – once sufficient budget is available, the system can allocate varying hint sizes across queries, and this flexibility quickly overtakes routing and cascading strategies. On GSM8K and HumanEval, Reactive Shepherding dominates across the entire cost range. We will add this analysis to the revised paper.
> > >
> > > **3.Example of failure:** We thank the reviewer for this question — it motivated us to closely examine cases where the LLM hint led the SLM astray, revealing insights that may help our algorithm design. We identified two distinct patterns:
> > >
> > > *Error propagation*: The LLM makes a subtle reasoning error: a double-subtraction (beehive problem), an off-by-one miscount (pizza slices), or a misreading of the problem setup (gift bags) — and the SLM anchors to the flawed hint, abandoning its own correct reasoning. For instance, in the pizza-slices problem, the SLM correctly identifies 5 people without a hint but switches to the LLM's incorrect count of 6 when the hint is provided.
> > >
> > > *Error amplification*: The LLM introduces a spurious step (multiplying by 24 hours in an electricity bill problem), and the SLM not only follows this error but over-extends it — the LLM answer is \\$504 (wrong), while the SLM under the bad hint produces \\$864, far worse than both the LLM's error and the correct answer of \\$21. This shows that bad hints can be worse than no hint at all.
> > >
> > > These cases are rare (affecting, for example,  0.4% queries in case of GSM8K after our predictor filtering) but they point to concrete directions for strengthening the shepherding policy. For example, incorporating a post-hint verification step that compares the SLM's hinted and unhinted answers, or detecting when the SLM's confidence drops after receiving a hint. We will include this analysis and discuss these design extensions in the revised paper

---

### Official Review · Reviewer_K432 · 2026-03-13

**Soundness:** 2
**Presentation:** 2
**Significance:** 2
**Originality:** 3
**Overall Recommendation:** 2
**Confidence:** 3

**Summary:**

This paper starts by setting up the problem of an “all-or-nothing” approach in traditional methods which invoke small and large language models together for improved quality. The proposed solution is LLM Shepherding, in which a hint from the large model is concatenated with the context and fed to the small model. The paper formalizes the problem of shepherding. Authors provide experiments comparing both reactive shepherding and proactive shepherding to baseline methods. There is a discussion as well regarding how much cost is being spent per response.

**Compliance With Llm Reviewing Policy:**

Affirmed.

**Final Justification:**

Based on my original comments, our discussion, and other reviews, I maintain my current score.

**Key Questions For Authors:**

Questions are listed above.

**Limitations:**

yes.

**Strengths And Weaknesses:**

Strengths
1. This is a novel idea and it seems as though a generalization of cascading or routing would be of interest to the community.
2. Reactive and proactive shepherding are compared with two other methods. The cost and accuracy are reported, as well as accuracy-cost efficiency. Furthermore, the performance is tested against four different datasets.

Weaknesses
1. It would be nice to exactly solve for the number of tokens needed as a hint, at least in theory (that is, Section 2 does not explicitly solve for this)
2. Corollary 2: The theory assumes that $c^{\text{in}} = c^{\text{out}} = 0$. While the result of Corollary 2 is nice, it does not imply that, for reduced cost, shepherding can also achieve quality $\tau$ as described in Eq. 2.
3. The policy as described in Section 3 is trained via a binary classifier and a regressor, rather than jointly training together. Little justification is provided for this reason. There is a vast literature on learning to defer which might justify this and is at least worth mentioning [1, 2]. Empirically, it would also be nice to validate this choice by perhaps training a different kind of router.
4. The description in Section 3.5 is very dense and overall it would help to cleanly write out the algorithm in a pseudocode environment. For example, what does it mean that the features are incorporated in training the shepherding model, and what is the motivation for this?
5. Empirically it seems as though, while the shepherding methods reduce costs, they do not perform well with respect to other methods in terms of accuracy (Tables 1, 2, 3, 4).

---

> ### Author Rebuttal · Authors · 2026-03-30
>
> We thank the reviewer for the constructive feedback. We address each point below.
>
> **1. Exact solution for minimum hint size.** Computing $n^{\*}(q)$ in closed form is intractable, as it depends on the query and on how SLM and LLM generate responses, which involve high-dimensional, non-convex generation processes with no analytical solution. Thus, we estimate $n^{\*}(q)$ via a learned two-stage predictor (Section 3).  This is analogous to the routing/cascading literature, where oracle decisions are defined implicitly (e.g., RouteLLM, FrugalGPT) and estimated empirically.
>
> **2. Corollary 2.2 and quality guarantee.** Corollary 2 does imply that Oracle shepherding achieves the quality threshold $\tau$ in (2). By definition (cf. (1)), $n^{\*}(q)$ is the minimum hint size required by the SLM to achieve the quality threshold $\tau$. Since Oracle shepherding uses $n^{\*}(q)$ for each query it satisfies the quality threshold while also achieving the least cost compared to routing and cascading strategies.
>
> **3. Two-stage design justification.** The hint-size distribution is zero-inflated (80.6% of GSM8K queries need no hint, with the remaining values following approximately exponential decay), the benefit of additional tokens is non-smooth, and the quality function is non-monotonic. A single regression model must handle both the dominant point mass at zero and the conditional distribution over positive hint sizes — and it fails on both counts. In our preliminary design using a single transformer-based regressor yielded mean absolute error of 42.8 on GSM8K, compared to 30.61 for the two-stage approach. Even when a query needs no hint, the single transformer-based regressor predicted a small non-zero value, triggering a spurious LLM API call that incurs the full input cost $|q| c_l^\text{in}$​. On GSM8K, where ~80% of queries are SLM-solvable, this inflated costs dramatically without accuracy benefit. In the two-stage approach, the binary classifier serves as a hard gate that eliminates unnecessary LLM calls entirely, while the conditional regressor focuses capacity on predicting hint sizes only for queries that need them..
>
> Regarding learning-to-defer: we appreciate the connection, but LLM shepherding has a fundamental difference. In learning-to-defer [Madras et al., 2018; Mozannar & Sontag, 2020], the "defer" decision hands the query entirely to a human/expert for the final response. In LLM shepherding, the "hint-needed" decision does not defer to the LLM for a complete response but only requests only a partial prefix, and the SLM generates the final answer. This distinction is fundamental to our cost advantage (Corollary 2.2). We will add these references and clarify the distinction in the revised manuscript.
>
> Madras, David, Toni Pitassi, and Richard Zemel. "Predict responsibly: improving fairness and accuracy by learning to defer." NeurIPS (2018).
>
> Mozannar, Hussein, and David Sontag. "Consistent estimators for learning to defer to an expert." ICML (2020).
>
> **4. Section 3.5 clarity and pseudocode.** We agree that Section 3.5 would benefit from clearer presentation. In the revised version, we will add pseudocode for both proactive and reactive shepherding.
>
> To clarify the reviewer's specific question: the SLM response features (average entropy, output length, query length) are concatenated with the text embedding and passed through the MLP to produce the fused representation. The motivation is that these features capture the SLM's epistemic uncertainty — higher entropy indicates disagreement across stochastic responses, signaling that a hint is likely needed. We validated this quantitatively via ablation on the CNK12 test set: removing entropy features caused the hint-needed binary classification F1 to drop by 5%, confirming that entropy is important for accurate routing decisions.
>
> **5. Accuracy vs. cost trade-off.** The accuracy differences in Tables 1–4 arise because each method operates at a different cost point, not because shepherding is less capable at a given budget. Methods like ABC achieve higher accuracy by spending more, not by being more efficient. Our Pareto cost-accuracy plots in the following link make this clear.
>
> *https://anonymous.4open.science/r/Pareto-0C04/*
>
> Reactive Shepherding's curve dominates the Pareto cost-accuracy curves across all datasets, achieving the highest accuracy at most cost levels and offering the optimal point on HumanEval, CNK12 and MBPP. For example, on GSM8K at $0.034, no baseline achieves higher accuracy than Reactive Shepherding's 89.1%. Furthermore, when accuracy is fixed at 90% of LLM accuracy (Figure 3, Section 4.3), Reactive Shepherding achieves the lowest cost across all benchmarks – 30% cheaper than ABC on GSM8K and 61% cheaper on MBPP. This confirms that shepherding delivers a genuinely superior cost-accuracy trade-off, not a cost reduction at the expense of accuracy.

---

> > ### Author Rebuttal · Reviewer_K432 · 2026-04-03
> >
> > Thank you to the authors for the detailed responses and clarifications. I appreciate the additional explanation, and I now understand that under the stated assumptions, Corollary 2.2 does also imply the quality guarantee.
> >
> > My main remaining concern is the assumption that the SLM has zero cost. As also discussed with Reviewers Z7Az and xoyF, this does not hold in many practical settings. While I understand the intended motivation, the current theory and experiments focus on this special case, and do not fully address the more general setting where the SLM incurs nonzero cost.
> >
> > Having read through other reviews, a second concern is about generality across tasks. From the discussion and appendix, it seems that hyperparameter tuning is needed at a task/dataset level. This does not invalidate the approach, but it does bring to mind the generality of the proposed method. In particular, it appears as though the hyperparameter choices are quite different across tasks in Table 5. Are the authors able to provide any insights into the extent to which the proposed method is brittle to hyperparameter choices?
> >
> > I also have one clarification question about the additional plots. For HumanEval and MBPP, could the authors explain why the final performance of Reactive Shepherding appears to exceed the large-only baseline? I am not sure how to interpret this in light of the discussion in KQ1 with Reviewer Z7Az, where it was noted that for very large hint sizes, the small model may still fail to recover the large model’s performance.

---

> > > ### Author Response · Authors · 2026-04-04
> > >
> > > We thank the reviewer for the follow-up questions and are glad our clarifications resolved the question on Corollary 2.2. We address the remaining concerns below.
> > >
> > > **Q1: SLM zero-cost.** SLM-LLM collaboration approaches, such as routing, cascading, and speculative decoding, are built on the premise that SLM inference costs are substantially lower than LLM inference costs. The larger this cost differential, the greater the gains in these approaches. Accordingly, the zero-cost SLM setting is widely adopted in the SLM-LLM collaboration literature:
> > >
> > > 1. Hybrid LLM [1] measures cost as the fraction of queries routed to the small model, with no cost attributed to SLM inference.
> > > 2. RouteLLM [2] measures cost as the percentage of calls to the strong model, excluding the weak model's cost entirely.
> > > 3. ABC [3] acknowledges SLM ensemble cost but excludes it from their edge-to-cloud cost accounting.
> > >
> > > Even when SLMs are accessed via paid APIs, the cost gap is typically substantial, as illustrated by representative providers below:
> > >
> > > | Provider | SLM| Input ($/1M) | Output ($/1M) | LLM| Input ($/1M) | Output ($/1M) |
> > > |-|-|-|-|-|-|-|
> > > | Artificial Analysis| Llama-3.2-3B | 0.08 | 0.08| Llama-3.3-70B | 0.58 | 0.71 |
> > > | Together AI | Gemma-3n-E4B | 0.02 | 0.04 | Llama-3.3-70B | 0.88 | 0.88 |
> > > | Groq | Llama-3.1-8B | 0.05 | 0.08 | Llama-3.3-70B | 0.59 | 0.79 |
> > >
> > > The SLM is 6-30× cheaper per token than the LLM and, in some cases, entirely free, reinforcing the zero-cost assumption as reasonable.
> > >
> > > Finally, we agree that nonzero SLM costs reduce the cost advantage of all cascading-based methods (including Reactive Shepherding, ABC, and FrugalGPT), as they invoke the SLM multiple times before escalation.This is not unique to the above approaches; it is inherent to all cascading-based methods. However, Proactive Shepherding is minimally affected: it makes the hint decision before SLM inference, so the SLM runs only once per query — the same as routing baselines — largely preserving its cost advantage.
> > >
> > > We will include a sensitivity analysis with nonzero SLM costs in the revised paper and report updated Tables 1-4 to transparently show how each method's advantage changes.
> > >
> > > **Q2.** *Sensitivity to hyperparameter choices.* Our method is not brittle to hyperparameter variation. The Pareto front curves demonstrate the performance of Reactive Shepherding across a wide range of hyperparameter configurations ($\eta_{\\text{hint}}$, K, $\alpha$). Across all four datasets, Reactive Shepherding consistently achieves favorable cost-accuracy trade-offs over a broad region of the hyperparameter space, rather than at a single narrow operating point.
> > >
> > > *Generality across tasks*. We acknowledge that Table 5 reports different hyperparameter values across datasets, but highlight three points.  First, this is standard practice across all baselines in our comparison: RouteLLM requires tuning its routing threshold, FrugalGPT requires tuning its confidence threshold, and GraphRouter switches between modes, as shown in Table 5. Our approach is no more demanding in this regard. Second, each hyperparameter serves an intuitive and distinct role: $\\alpha$ controls the cost-accuracy trade-off, $\\eta_{\\text{hint}}$ determines when a query is too complex for hinting and should go directly to the LLM, and K governs the confidence level of the cascading decision. Third, and most importantly, our model itself generalizes across tasks without retraining. For example, the HumanEval and MBPP results in Tables 3-4 are obtained using the model trained solely on GSM8K. This demonstrates that the learned hint allocation captures task-general patterns of query difficulty, and only lightweight hyperparameter calibration (not model retraining) is needed for new tasks.
> > >
> > > **Q3: Reactive Shepherding exceeding LLM-only baseline on HumanEval and MBPP.** We thank the reviewer for this keen observation. This occurs because Reactive Shepherding's cascading mechanism introduces an additional source of correct answers that the LLM-only baseline lacks. When the SLM achieves majority agreement across three samples (N=2 per Table 5), its consensus output is used directly without invoking the LLM. On code generation tasks, the SLM's majority-voted solution is sometimes correct on queries where the LLM's single-pass output fails to compile or contains bugs. The system thus exceeds the LLM-only baseline by combining two complementary strengths: SLM consensus for queries where agreement signals reliability, and LLM hints for harder queries. We observed the same phenomenon with ABC, which also uses SLM agreement as a gating mechanism. We will add this explanation when we add these new figures to the revised paper.
> > >
> > > [1]Ding, Dujian, et al. "Hybrid llm: Cost-efficient and quality-aware query routing." ICLR (2024)
> > >
> > > [2]Ong, Isaac, et al. "Routellm: Learning to route llms with preference data." ICLR (2025)
> > >
> > > [3]Kolawole, Steven, et al. "Agreement-based cascading for efficient inference." TMLR(2025)

---

### Decision · Program_Chairs · 2026-04-30

**Decision:**

Reject

**Comment:**

The paper proposes a method for combining small language models (SLMs) and large language models (LLMs) to reduce the costs of running large scale models. The core idea of the paper is to request only a few tokens from the LLM to provide a "hint" that will guide the SLM, thus significantly improving the performance of the SLM at a minimal cost.

The reviewers noted that the problem studied by the paper is important and timely. Reviewers mentioned that the core idea is novel and well-motivated, and that the strategy and framework proposed in the paper are reasonable and results are overall valid. However, some reviewers noted that the cost-analysis presented in the paper appears to be misleading, attributing zero-cost to the SLM, where in fact the SLM cost should be taken into account. After further discussion with the reviewers, it seems that all reviewers believe that ignoring the cost of the SLM may have inflated the numbers reported in the tables summarizing the main experimental results. Reviewers were thus worried that the results presented in the paper do not accurately capture the true gains from the proposed method.

I therefore lean to reject the paper, and encourage the authors to fix the issues pointed out by the reviewers, particularly to take into account the cost of the SLM in order to provide a more honest and accurate summary of the results.